# Control of telomere length in yeast by SUMOylated PCNA and the Elg1 PCNA unloader

Pragyan Singh, Inbal Gazy, Martin Kupiec*

The Shmunis School of Biomedicine and Cancer Research, The George S. Wise Faculty of Life Sciences, Tel Aviv University, Tel Aviv, Israel

**Abstract** Telomeres cap and protect the linear eukaryotic chromosomes. Telomere length is determined by an equilibrium between positive and negative regulators of telomerase activity. A systematic screen for yeast mutants that affect telomere length maintenance in the yeast *Saccharomyces cerevisiae* revealed that mutations in any of ~500 genes affects telomere length. One of the genes that, when mutated, causes telomere elongation is *ELG1*, which encodes an unloader of PCNA, the processivity factor for replicative DNA polymerases. PCNA can undergo SUMOylation on two conserved residues, K164 and K127, or ubiquitination at lysine 164. These modifications have already been implicated in genome stability processes. We report that SUMOylated PCNA acts as a signal that positively regulates telomerase activity. We also uncovered physical interactions between Elg1 and the CST (Cdc13-Stn1-Ten) complex and addressed the mechanism by which Elg1 and Stn1 negatively regulates telomere elongation, coordinated by SUMO. We discuss these results with respect to how chromosomal replication and telomere elongation are coordinated.

*For correspondence:
martin@tauex.tau.ac.il

Competing interest: The authors declare that no competing interests exist.

## eLife assessment

This **important** study aims to discover the mechanisms governing the switch between conventional DNA replication and the specialized mechanism of telomere end replication. **Solid** genetic and biochemical assays suggest an interplay between sumoylated PCNA and chromosome terminal capping proteins. The questions addressed have implications for several fields, such as genome stability.

## Introduction

Telomeres cap the ends of linear eukaryotic chromosomes and safeguard them against genome instability. Telomeric DNA consists of G/C-rich DNA repeats, with the G-rich strand extending to form a 3' single-stranded overhang. Telomeres are primarily involved in counteracting the gradual shortening of the telomeric DNA caused by conventional DNA replication, thereby solving the 'end-replication problem' (*Bonnell et al., 2021*; *Soudet et al., 2014*). Another important function of telomeres is to protect chromosome ends from improper recognition as a DNA double-strand break (DSB) (reviewed in *de Lange, 2018*).

An important, conserved protein complex involved in the regulation of telomere length is the CST, which in *Saccharomyces cerevisiae* is composed of Cdc13, Stn1, and Ten1 (*Chen and Lingner, 2013*; *Ge et al., 2020*; *Lim et al., 2020*; *Puglisi et al., 2008*; *Rice and Skordalakes, 2016*). The CST has similarity to the ssDNA binding complex RPA and localizes specifically to the single-stranded telomeric DNA, where it is involved in chromosome end capping and telomere length regulation (*Hughes et al., 2000*; *Lin and Zakian, 1996*; *Mersaoui and Wellinger, 2019*). The CST facilitates

telomerase-mediated telomere elongation (*Evans and Lundblad, 1999*; *Evans and Lundblad, 2000*; *Nugent et al., 1996*), participates in telomere processing during DNA replication (*Soudet et al., 2014*), and helps 'capping' the telomere (*Mersaoui and Wellinger, 2019*). In late S-phase, Cdc13 interacts with the Est1 subunit of telomerase and promotes telomerase activity (*Chandra et al., 2001*; *Chen et al., 2018*). Stn1 binds Cdc13 in a region that overlaps with that bound by Est1. Recruitment of Stn1 evicts Est1 and prevents further elongation of the G-strand (*Chandra et al., 2001*; *Wang et al., 2000*). The amino terminus of Stn1 binds Ten1, whereas its C-terminus interacts with both Cdc13 and Pol12, a subunit of Polymerase Alpha (*Grossi et al., 2004*; *Petreaca et al., 2006*; *Puglisi et al., 2008*). Thus, the CST coordinates leading strand elongation by telomerase with lagging strand DNA synthesis by the cell's replisome (*Grossi et al., 2004*).

Very often the integrity of the genome is compromised by internal and external sources of DNA damage. This vulnerability increases during S-phase when the DNA is unpacked and copied. Elg1, the major subunit of a replication factor C-like complex (RLC), plays a central role in DNA replication and is critical for genome maintenance. In *S. cerevisiae*, loss of the *ELG1* gene causes gross chromosomal rearrangements, chromosome losses, defective sister chromatid cohesion and recombination, increased sensitivity to DNA damaging agents, and abnormal telomere length maintenance (reviewed in *Arbel et al., 2021*). The function of the Elg1 RLC is to unload the processivity factor PCNA from chromatin, in particular after Okazaki fragment processing and ligation in the lagging strand, and during DNA repair (*Kubota et al., 2015*; *Kubota et al., 2013*; *Parnas et al., 2010*; *Shemesh et al., 2017*). The mammalian ortholog of *ELG1* (ATAD5) participates in the Fanconi Anemia pathway (*Kim et al., 2020*; *Yang et al., 2011*) and, when mutated, leads to genome instability and cancer in mice and humans (*Bell et al., 2011*; *Kuchenbaecker et al., 2015*; *Maleva Kostovska et al., 2016*).

Post-translational modifications of PCNA (in particular ubiquitination and SUMOylation) orchestrate the activity of a large number of interacting proteins during DNA replication and repair. Many of these proteins interact through a PIP (PCNA Interacting Peptide) motif. PCNA can undergo SUMOylation on two conserved residues, K164 and K127, or ubiquitination at lysine 164. Although the Elg1 RLC shows affinity to SUMOylated PCNA, the complex is able to unload modified and unmodified versions of the clamp (*Kang et al., 2019*; *Kubota et al., 2015*; *Kubota et al., 2013*; *Parnas et al., 2010*; *Shemesh et al., 2017*; *Shiomi and Nishitani, 2013*). Thus SUMOylation may assist but is not essential for PCNA unloading. Retention and accumulation of PCNA on DNA is the major cause of genome instability in *elg1Δ* (*Itzkovich et al., 2023*; *Johnson et al., 2016*; *Shemesh et al., 2017*).

Previous studies found that telomeric proteins, and in particular the CST, can be regulated by SUMO modification (*Hang et al., 2011*). Here we show that SUMOylation of PCNA plays a role in the regulation of telomere length and uncovers an interaction between the Elg1 RLC and the CST. We report that PCNA SUMOylation positively regulates telomerase activity, and its unloading by the Elg1-RLC is essential for normal telomere size regulation. The N-terminus of Elg1 interacts with Stn1 during late S-phase and mediates the interaction between Stn1 and Cdc13 after PCNA has been unloaded from the telomeres.

## Results

### The long telomeres phenotype of *elg1Δ* strains requires PCNA SUMOylation

Previous results suggested that the Elg1 protein preferentially interacts with the SUMO machinery and with SUMOylated PCNA (*Parnas et al., 2011*; *Parnas et al., 2010*; *Shemesh et al., 2017*). To test whether PCNA modifications play any role in the elongated telomere phenotype of *elg1Δ*, we combined the *ELG1* deletion with mutations in the *POL30* gene, encoding PCNA. We mutated either lysine 127, lysine 164, or both (hereafter referred as *pol30-K127R*, *pol30-K164R*, and *pol30-RR*, respectively) in the *elg1Δ* background. *Figure 1* shows that the mutations in *POL30* have little effect by themselves. Importantly, whereas the single mutants did not affect the long telomeres of *elg1Δ*, mutating both lysine residues of each PCNA subunit completely abrogated the long telomere phenotype. Lysine 164 can be modified by both SUMO and ubiquitin; mutating only this residue prevents ubiquitination, but not SUMOylation. *Figure 1* demonstrates that this mutant shows no effect. To ensure that SUMO, and not ubiquitin, is the modification responsible for the long telomeres of *elg1Δ*, we deleted *RAD18*, encoding the E3 ubiquitin ligase required to ubiquitinate PCNA (*Hoege et al.,*

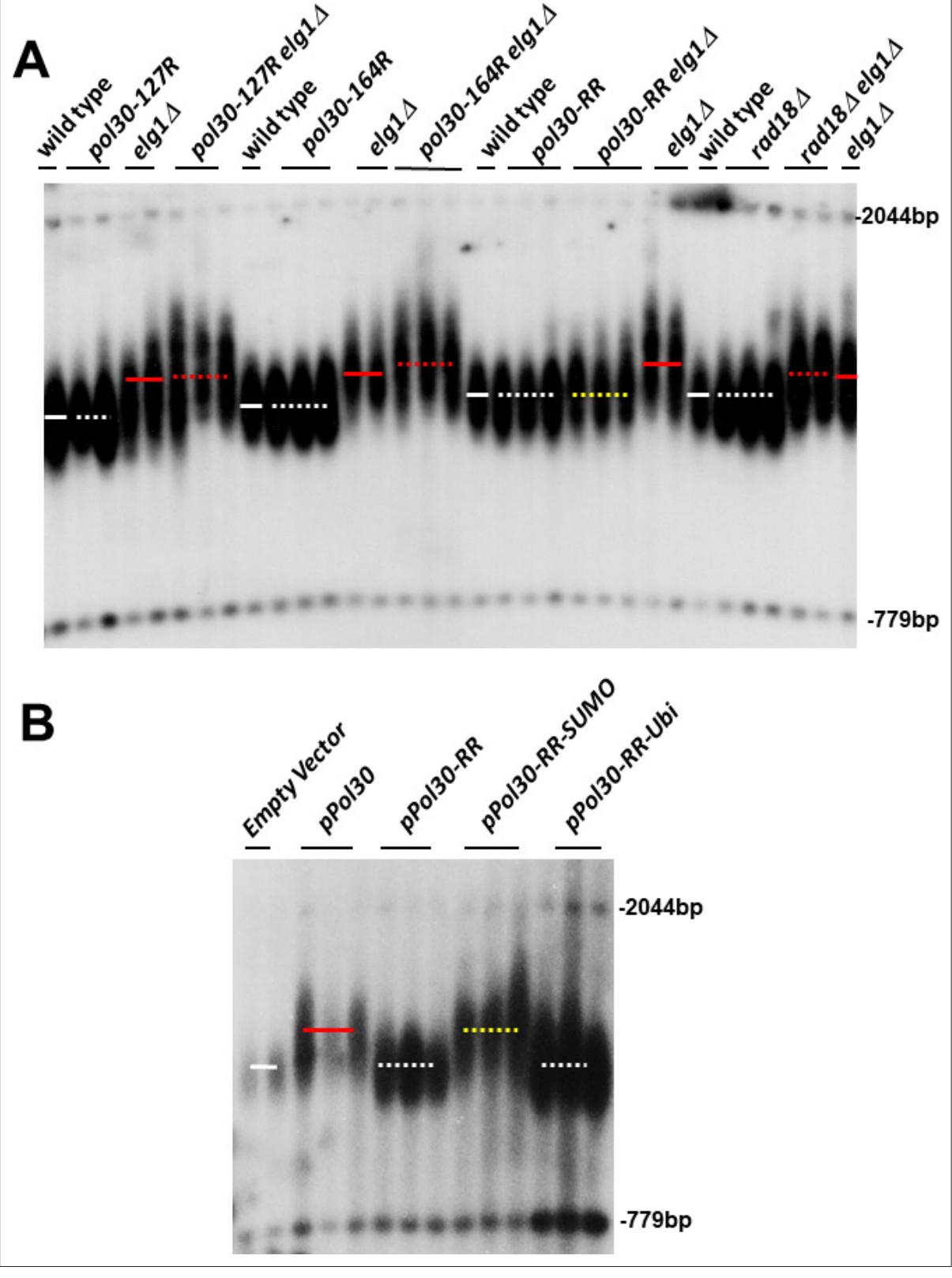

**Figure 1.** SUMOylated PCNA regulates telomere length. (**A**) Southern blot (Teloblot) showing that lack of SUMOylation of PCNA prevents telomere elongation. Independently created colonies were passaged ten times, its DNA extracted, digested with *Xho*I, and run in an agarose gel. The DNA was then transferred to a nitrocellulose membrane, which was incubated with a radioactive probe that detects telomeres, and a size marker. (**B**) Teloblot showing that overexpression of wild-type PCNA or Pol30-RR-SUMO fusion, but not Pol30-ubiquitin fusion or Pol30-RR causes telomere elongation.

*Figure 1 continued on next page*

*Figure 1 continued*

The online version of this article includes the following source data for figure 1:

**Source data 1.** Source data is the original Southern blot.

*2002*; *Stelter and Ulrich, 2003*). *Figure 1* shows that lack of Rad18 has no effect on the elongated telomeres of *elg1Δ*. (Deletion of the SUMO E3 ligases *SIZ1* and *SIZ2* [to prevent SUMOylation] by itself alters telomere length [*Hang et al., 2011*] and thus they could not be used here.) We conclude that SUMOylation of PCNA is required for the long telomeres phenotype of *elg1Δ* mutants.

These results suggest that SUMOylated PCNA may play a central role in determining telomere length also in cells with wild-type Elg1 activity. We thus next asked whether artificially overexpressing modified and unmodified PCNA may mimic the long telomere phenotype of *elg1Δ*. Since it is hard to force ubiquitination or SUMOylation of PCNA in vivo, we used covalent fusions between PCNA and either ubiquitin or SUMO. These have been shown in the past to mimic naturally modified PCNA molecules (*Parker et al., 2007*; *Takahashi et al., 2020*). We overexpressed either wild-type Pol30, Pol30-RR (which cannot be modified) ,or Pol30-RR fused either to ubiquitin or to SUMO, in *pol30-RR* cells, unable to carry out modifications of the PCNA encoded by the genome. *Figure 1B* shows that telomeres become elongated upon overexpression of the wild-type protein (which can be modified), or the Pol30-RR version fused to SUMO, but do not elongate upon overexpression of the unmodifiable Pol30-RR version, or the one fused to ubiquitin. These results, together with those of *Figure 1A*, show that accumulation of SUMOylated PCNA, and not unmodified or ubiquitinated PCNA, is both necessary and sufficient for telomere elongation in the presence or in the absence of Elg1.

## Genetic and physical interactions between Elg1 and Stn1

The elongated telomere phenotype of Elg1 was identified in systematic screens of the yeast knockout collection (*Askree et al., 2004*; *Gatbonton et al., 2006*), together with ~200 additional mutants of similar phenotypes. To identify the pathway of telomere length maintenance in which Elg1 participates, we systematically combined the *elg1Δ* mutant with all the other mutants with long telomeres. Most double mutants exhibited telomeres longer than each of the single mutants, indicating that they affect different pathways. One of the mutants showing clear epistasis (the double mutant was not longer than the single mutants) carried a deletion of *RRP8*, a gene involved in modification of the ribosomal RNA. Further analysis demonstrated that the knockout of *RRP8* exhibited the 'neighboring gene effect,' and in fact the deletion of *RRP8* caused a decrease in the expression of its neighboring gene, *STN1* (*Ben-Shitrit et al., 2012*). We confirmed the genetic interactions between *ELG1* and *STN1* alleles by combining *elg1Δ* with the *stn1-13* and *stn1-164* alleles (*Grandin et al., 1997*), which exhibit long telomeres. The telomeres of the double mutants are not longer than those of the single mutants (*Figure 2A*), indicating that Elg1 works in the same TLM pathway as Stn1.

Given the epistatic relations between the *ELG1* and *STN1* genes, we addressed, by co-immunoprecipitation (IP) and yeast two-hybrid (YTH) assay, whether the proteins in the CST complex interact with Elg1. IP of epitope-tagged Elg1 pulled down Stn1 but not the other two members of the CST complex, Cdc13 or Ten1 (*Figure 2B*). For our YTH experiments, the Elg1 protein was divided into three main functional domains: an N-terminal domain (amino acids 1–234), a central AAA domain (amino acids 235–514), and a C-terminal domain (amino acids 515–791) (*Itzkovich et al., 2023*; *Figure 3A*). The AAA and C-terminal domains contain the conserved RFC boxes and are required for the interactions with the small RFC subunits, shared by all RFC-like domains. The N-terminal domain (hereafter referred to as NTD) is unique to Elg1 (*Arbel et al., 2021*). We thus concentrated on this domain and used the C-terminus as a negative control as it does not interact with any of the proteins tested here.

The NTD contains SUMO-interacting motifs (SIMs), which were previously shown to interact with SUMO and the SUMO machinery (*Parnas et al., 2011*; *Parnas et al., 2010*; *Figure 3A*). We used the YTH technique to test potential interactions between Elg1 and the CST components, and their dependency on SUMO. The N-terminus of Elg1 interacted strongly with Cdc13 and Stn1, and weakly with Ten1 (*Figure 3B, C*, *Figure 3—figure supplement 1*). In contrast, no interaction was detected with Elg1's CTD. We could also detect interactions of Cdc13, Stn1, and Elg1's NTD with SUMO (*Figure 3*). These were used as positive controls in all our experiments. Since the results with Ten1 were much weaker than those with Cdc13 and Stn1, we concentrated on these two last proteins. Below, we

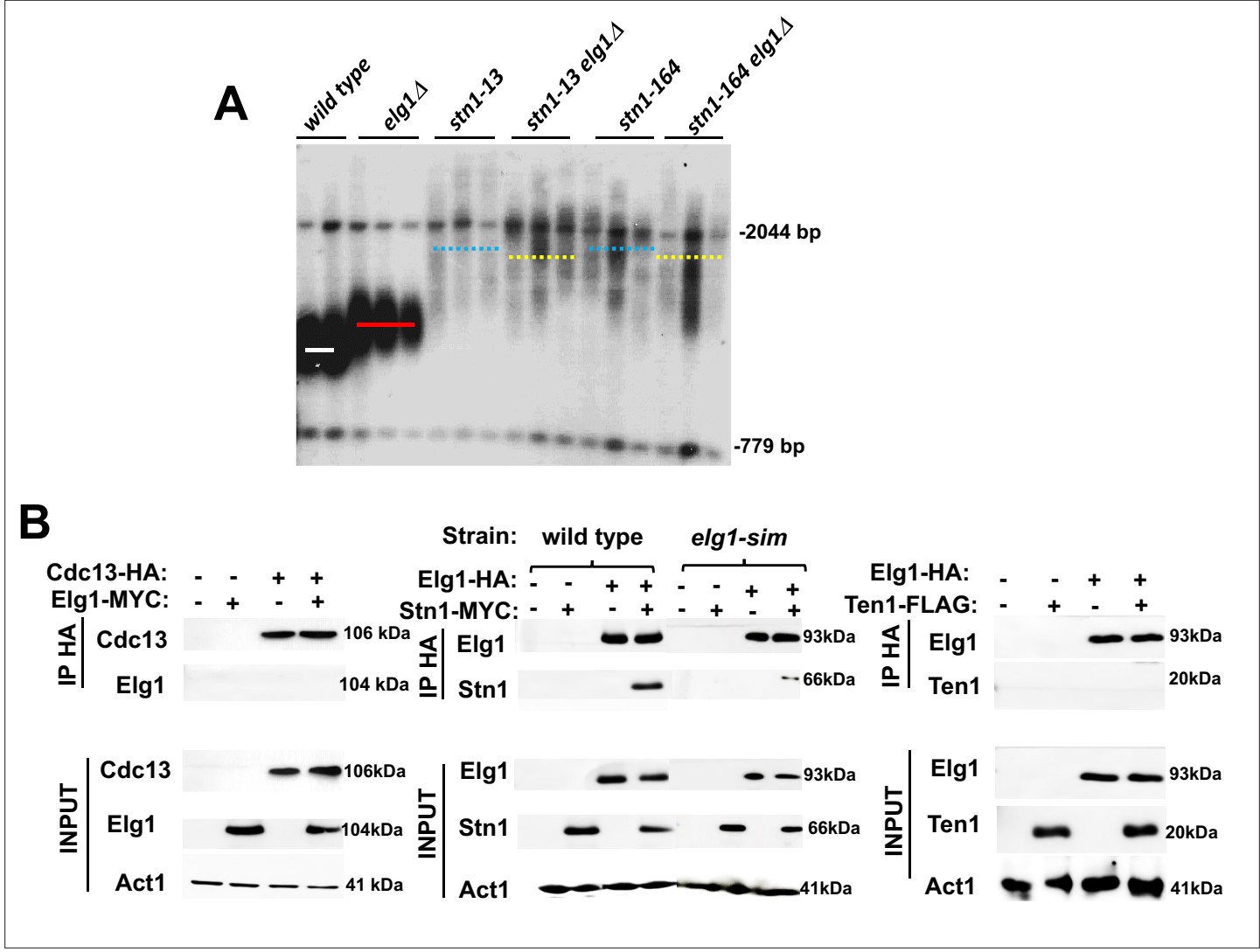

**Figure 2.** Genetic and physical interaction between *ELG1* and *STN1*. (**A**) Teloblot showing epistasis between *elg1Δ* and *stn1* mutants. (**B**) Co-Immunoprecipitation experiment showing physical interaction between Elg1 and Stn1 and reduced physical interaction between Elg1-sim and Stn1. No interaction could be detected between Elg1 and Cdc13 or Ten1.

The online version of this article includes the following source data for figure 2:

**Source data 1.** The source data contains the Southern blots and the Western blots used to make the figure.

dissect, using YTH in various genetic backgrounds, the interactions between Elg1, Stn1, Cdc13, and PCNA. The results are also summarized in *Table 1*.

Interestingly, the interaction of Elg1's NTD with Stn1, but not with Cdc13, was abolished when the SIM motifs (amino acids I27A, I93K, I121A, and I122A) were mutated (*Figure 3B and C*). We confirmed these results by co-immunoprecipitation (co-IP) (*Figure 2B*). These results suggest that the interaction with Stn1 is mediated by SUMOylation of a protein, whereas the interaction of Cdc13 with Elg1-NTD is independent of the SIM in the plasmid-borne copy of Elg1. We confirmed this result by deleting the genes encoding the SUMO-specific E3 enzymes, *SIZ1* and *SIZ2* in the genome (a double deletion is necessary, as many times each protein can compensate for the lack of the other). *Figure 3D* shows that indeed deletion of the genes encoding these enzymes abolishes the interaction of Stn1 and Cdc13 with Elg1's NTD. We conclude that SUMO plays a role in mediating the Elg1-CST interactions. Since Elg1 preferentially interacts with SUMOylated PCNA (*Parnas et al., 2010*) and Stn1 can bind SUMO noncovalently in YTH (*Figure 3C*), we reckoned that SUMOylated PCNA may mediate the interaction between Stn1 and Elg1. We therefore tested whether abolishing PCNA's SUMOylation

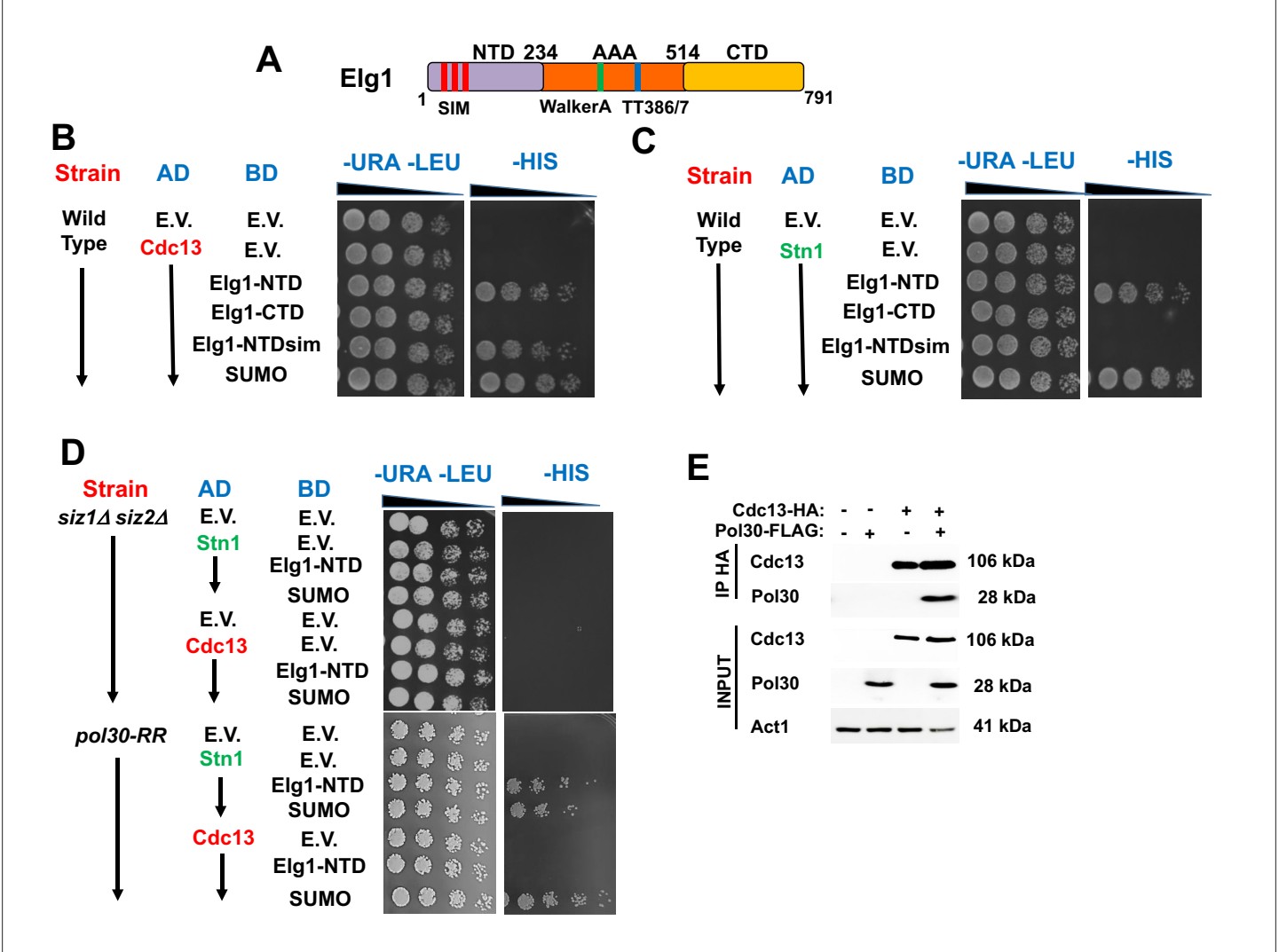

**Figure 3.** Determinants of the Elg1-Stn1 interaction. (**A**) Schematic representation of the Elg1 protein. The three SIM motifs, the WalkerA motif and the two threonines at the interface with PCNA, are shown. (**B**) Yeast two-hybrid (YTH) interaction of Cdc13 and Elg1 in a wild-type strain. AD: protein fused to the activating domain of Gal4; BD: protein fused to the DNA binding domain of Gal4. e.v.: empty vector. (**C**) YTH interaction of Stn1 and Elg1 in a wild-type strain. (**D**) YTH experiments in the *siz1Δ siz2Δ* and *pol30-RR* background. (**E**) Co-Immunoprecipitation experiment showing physical interaction between Cdc13 and PCNA.

The online version of this article includes the following source data and figure supplement(s) for figure 3:

**Source data 1.** The Source data contains the Western blots used in *Figure 3E*.

**Figure supplement 1.** Weak interaction between Elg1 and Ten1.

**Figure supplement 2.** Cdc13 interacts with PCNA, and mutations that prevent PCNA SUMOylation (*pol30-RR*) also impair its interaction with Cdc13.

sites in the genome (*pol30-RR*) had an effect on the Stn1-Elg1 interaction. We found, however, that Elg1 and Stn1 could still interact in the *pol30-RR* strain (*Figure 3D*), despite the fact that PCNA cannot become SUMOylated. Consequently, a different SUMOylated target seems to mediate the interaction between Elg1 and Stn1, or else either one of the proteins is SUMOylated. Despite numerous attempts, we were unsuccessful in observing SUMOylation of either Stn1 or Elg1.

We also noted that, despite the fact that mutation in the plasmid-borne NTD's SIM had no effect on the interaction with Cdc13 (*Figure 3B*), deletion of *SIZ1* and *SIZ2*, and mutation in the SUMOylation sites in PCNA completely abolished the interactions between Elg1's NTD and Cdc13 (*Figure 3D*). This implies that PCNA SUMOylation is necessary for the interaction between Elg1 and Cdc13. We confirmed by co-IP that Cdc13 indeed physically interacts with PCNA (*Figure 3E*). The interaction

**Table 1.** Summary of all yeast two-hybrid (YTH) interactions presented.

| Strain | Plasmid | Interaction w/Stn1 | Interaction w/Cdc13 |
|---|---|---|---|
| Wild type | Elg1-NTD | Yes | Yes |
| Wild type | Elg1-sim | No | Yes |
| *siz1Δ siz2Δ* | Elg1-NTD | No | No |
| *pol30-RR* | Elg1-NTD | Yes | No |
| *elg1Δ* | Elg1-NTD | Yes | No |
| *elg1-sim* | Elg1-NTD | Yes | Yes |
| *elg1-DD* | Elg1-NTD | Yes | No |
| *elg1-sim+DD* | Elg1-NTD | Yes | No |
| *elg1-WalkerA* | Elg1-NTD | Yes | No |
| *cdc13-snm* | Elg1-NTD | Yes | Yes |
| Wild type | Elg1-NTD | - | *snm:* no |

elg1-sim: mutation in the SUMO-interacting motif of Elg1; *elg1-DD: TT386/7DD; elg1-sim+DD*: combination of mutations in the SIM and in TT386/7; *elg1-WalkerA*: mutation that eliminates ATPase and unloading activity of Elg1. *cdc13-snm:* allele of Cdc13 that cannot be SUMOylated.

between Cdc13 and PCNA was almost completely abolished when PCNA could not be SUMOylated (*Figure 3—figure supplement 2*), In summary, the interaction between Cdc13 and both PCNA and Elg1, but not that of Elg1 with Stn1, is dependent on SUMOylation of PCNA.

## Elg1's functional activity is essential for its interaction with Cdc13

When we repeated the YTH assays in a strain in which the *ELG1* gene is deleted from the genome, we were surprised to see that now Cdc13 failed to interact with the plasmid-borne Elg1-NTD (*Figure 4A*, compare to *Figure 3B*). In contrast, the interaction between Stn1 and Elg1-NTD was not affected by the deletion of the genomic *ELG1*.

Taken together, these results imply that whereas Stn1 interacts directly with the plasmid-borne Elg1-NTD, via its SIM-mediated SUMO-binding, the interaction with Cdc13 require both PCNA SUMOylation and the genomic *ELG1* gene. We can envision two formal possibilities for the latter requirement: either (1) the interaction with Cdc13 requires parts of the Elg1 protein not present in the plasmid-borne NTD (e.g., the AAA or CTD domain) or (2) Elg1's activity (e.g., PCNA unloading) may be required to facilitate the physical interaction of the NTD with Cdc13.

To distinguish between these two possibilities, we carried out YTH experiments with strains carrying different alleles of *ELG1* in the genome. Three mutations were tested first: *elg1-sim* (mutated in amino acids I27A, I93K, I121A I122A), which does not interact with SUMO; *elg1-TT386/387DD*, hereafter referred to as *elg1-DD*, which affects the interface Elg1-PCNA (*Shemesh et al., 2017*); and *elg1-sim+DD*, an allele that combines both mutations. Whereas the *SIM* mutation has a very mild effect on the ability of cells to unload PCNA, and displays no phenotypes, the *elg1-DD* is more impaired and has a stronger phenotype, and the *elg1-sim+DD* allele has essentially a null phenotype, similar to that of *elg1Δ*, but expressing the protein at normal levels (*Itzkovich et al., 2023*; *Shemesh et al., 2017*).

We found that in strains carrying genomic *elg1-DD* and *elg1-sim+DD* alleles, Cdc13 failed to interact with plasmid borne Elg1-NTD, whereas in the *elg1-sim* strain the interaction was weak but clearly seen (*Figure 4B–D*). This observation indicates that the PCNA unloading activity of Elg1 is essential to allow the interaction between Cdc13 and the plasmid borne Elg1-NTD: this activity is high in *elg1-sim* mutants, intermediate in the *elg1-DD* strain, and is abolished by the *elg1-sim+DD* mutations. Note that the presence of a functional SIM at the plasmid-borne NTD is sufficient to ensure a mild interaction with Cdc13, but only if the genomic copy retains PCNA unloading activity (as in the *elg1-sim* mutant). On the other hand, the presence of a functional SIM in the inactive *elg1-DD* allele does not warrant an interaction of Elg1's NTD (which has SIM motifs) with Cdc13. We confirmed this idea by analyzing the interaction between Cdc13 and Elg1's NTD in another SIM-containing mutant

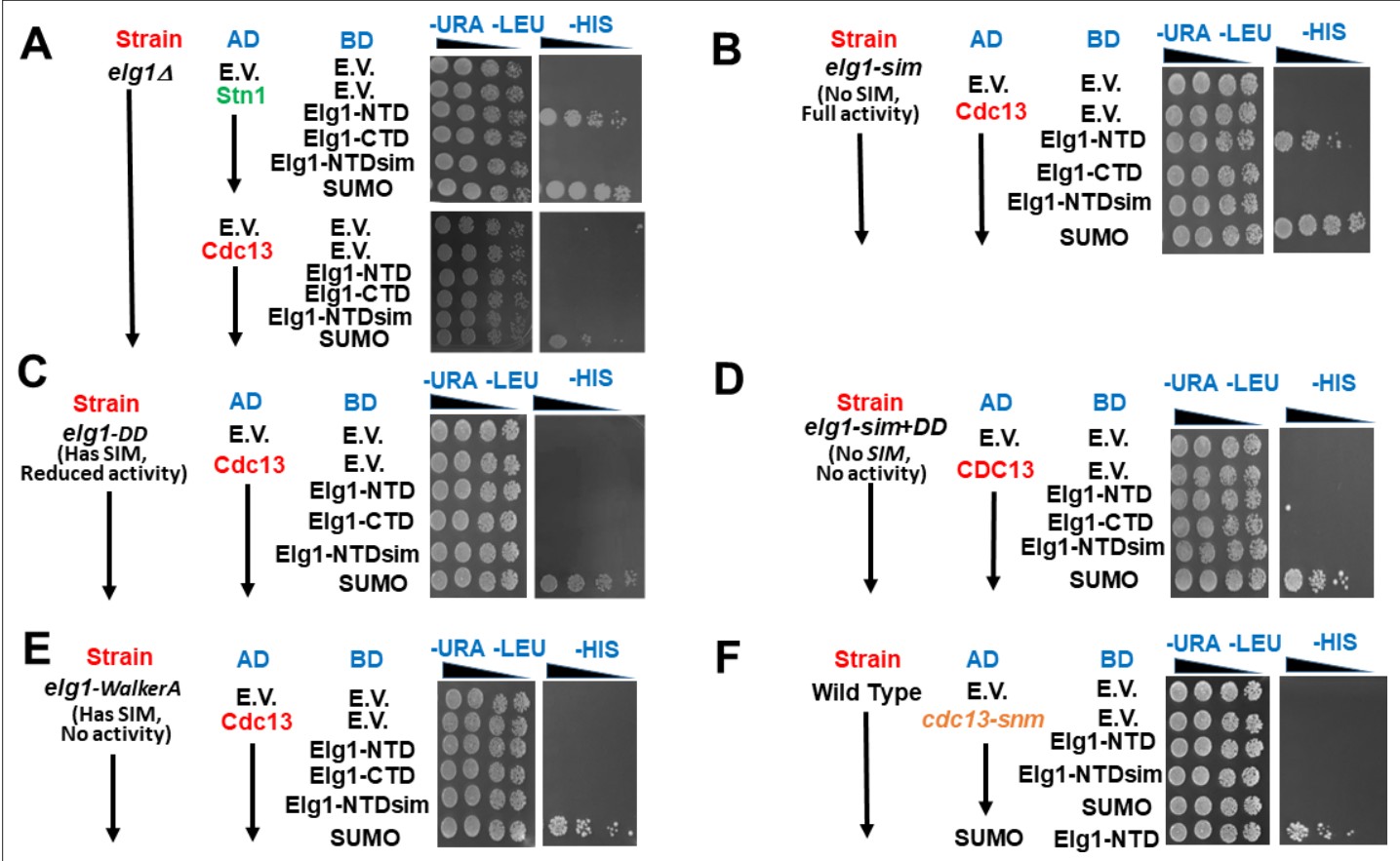

**Figure 4.** The unloading activity of Elg1 and Cdc13 SUMOylation are necessary for the Elg1-Cdc13 interaction. (**A**) Yeast-two-hybrid (YTH) experiment in a *elg1Δ* strain. (**B**) YTH experiment in a *elg1-sim* strain. (**C**) YTH experiment in a *elg1-TT386/7DD* strain. (**D**) YTH experiment in a *elg1-sim+DD* strain. (**E**) YTH experiment in a *elg1-Walker AB* strain. (**F**) Lack of interaction between *cdc13-snm* and Elg1 (in a wild-type strain).

The online version of this article includes the following source data and figure supplement(s) for figure 4:

**Figure supplement 1.** All *elg1* alleles used are expressed at similar levels.

**Figure supplement 1—source data 1.** The Source Data contains the Western Blot used in the figure.

**Figure supplement 2.** SUMOylation of the genomic copy of Cdc13 has no effect on the interactions of a wt copy with Stn1 or Elg1.

strain devoid of PCNA unloading activity, *elg1-WalkerA* (KK343/4DD), which lacks ATPase and PCNA unloading activity (*Itzkovich et al., 2023*). This mutant expresses wild-type levels of protein, and the mutant has essentially the same phenotype of an *elg1Δ* allele. *Figure 4E* shows that no interaction between the plasmid-borne Elg1 NTD and Cdc13 could be detected despite the fact that functional SIMs are present both at the NTD and in the genome-encoded Elg1-WalkerA protein. All the mutants expressed the Elg1 protein at wild-type levels (*Figure 4—figure supplement 1*).

We conclude that whereas the interaction with Stn1 is direct and mediated by the SUMOylation of a protein (different from PCNA), the interaction between Elg1's NTD and Cdc13 is indirect and can only take place after PCNA has been SUMOylated and unloaded. This suggests a model for the coordination of telomerase activity and DNA replication: the presence of SUMOylated PCNA at telomeres may serve as a positive signal for telomerase activity. Once Elg1 unloads PCNA, however, Stn1-Elg1 can bind Cdc13, ending telomerase activity. In this context, it is possible to understand the results seen in *Figure 3D*, showing that mutations in the SUMO E3 enzymes or in the sites of SUMOylation of PCNA, both of which prevent PCNA SUMOylation, show the same effect as the *sim+DD* or *walkerA* mutations in the genomic *ELG1* copy, which prevent PCNA unloading activity.

## SUMOylation of Cdc13 is needed for interaction with Elg1

Cdc13 undergoes SUMOylation, and this modification plays an essential role in the negative regulation of telomere length (*Hang et al., 2011*). Thus, Cdc13 is a good candidate for the target of SUMOylation required for the interaction between Elg1 and Stn1. Accordingly, mutations of the Cdc13 SUMOylation sites (*cdc13-snm*, 'SUMO no more') weaken its interaction with Stn1 and lead to elongated telomere phenotype (*Hang et al., 2011*). We therefore checked whether the *cdc13-snm* mutation affects also the interaction with Elg1. *Figure 4F* shows that indeed preventing SUMOylation of Cdc13 abolishes the interaction between this protein and Elg1. Note that the strain used expresses a wild type genomic copy of *CDC13*. Thus, SUMOylation of the plasmid-borne version of Cdc13 (i.e., in *cis*) is required for these interactions in the YTH assay. No effect was observed when the YTH plasmid carried a wild-type copy of *CDC13*, and the genomic copy was the *cdc13-snm* allele (*Figure 4—figure supplement 2A*). In summary, the N-terminus of Elg1 interacts with Cdc13 only if (1) Cdc13 can be SUMOylated (*Figure 4F*), (2) PCNA can be SUMOylated (*Figure 3D*), and (3) PCNA can be unloaded (*Figure 4A, C–E*).

## The interaction of Elg1 with Stn1 takes place only at late S-phase

The fact that the interactions of the Elg1 NTD with Cdc13 are dependent on PCNA modification and unloading, whereas those with Stn1 are not, suggests that the interaction with Cdc13 may be mediated by, or dependent on, the interaction of Elg1 with Stn1. To further dissect this point, we divided the Stn1 protein into an NTD (first 281 amino acids) and a CTD (amino acids 282–495). The later has been shown to be the region that interacts with Cdc13 (*Petreaca et al., 2007*). *Figure 4—figure supplement 2B* shows that the NTD of Elg1 also interacts with Stn1 via its C-terminal domain, both in *WT* and *elg1Δ* strains.

Having established Stn1 as the main interactor of Elg1 in the CST complex, we next examined the interaction between these proteins throughout the cell cycle phases. We arrested a strain with tagged Elg1, Stn1, and DNA Polymerase Delta in G1 with alpha factor. Cells were released into the cell cycle and samples were taken at intervals during two cell cycles. This allowed us to map the timing of interaction between Elg1 and Stn1 proteins, and whether this interaction is codependent on the movement of replication fork. *Figure 5—figure supplement 1* shows that the total level of these proteins does not change throughout the cell cycle. We immunoprecipitated Elg1 and measured the level of the other two proteins by western blot (*Figure 5A–C*). Pol3 was co-IPed with Elg1 strongly only during the S-phase (40–70 min, 120–140 min), whereas Stn1 was only detected in late S-phase (60–70 min, 140–160 min) for both cell cycles and was barely detected in G1 (*Figure 5A–C*). This pattern coincides with telomerase activity at telomeres, which is low in early to mid S-phase and peaks in late S-phase (*Puglisi et al., 2008*; *Taggart et al., 2002*).

To monitor the arrival of the replication fork to telomeres, we performed chromatin immunoprecipitation at telomeres (Telo-ChIP) in synchronized cells. *Figure 5D* shows that the timing of arrival of PCNA to the telomeres (60–70 min into the cell cycle) coincides with the timing of the co-IP between Elg1 and Stn1 (*Figure 5A–C*). Thus, our results are consistent with the idea that Elg1 moves with the replication fork, and interacts with Stn1 at the end of S-phase, when telomeres are replicated and telomerase is activated.

## The interaction between Cdc13 and Stn1 is dependent on Elg1

Our YTH and IP data suggest that unlike the relationship between Cdc13 and Elg1, the interaction between Stn1 and Elg1 is direct (*Figures 2 and 3*). Since both Elg1 and Cdc13 bind Stn1 at its C-terminus (*Figure 4—figure supplement 1* and *Petreaca et al., 2007*), it is possible that the absence of Elg1 may affect the interaction between the two CST members. We thus measured the co-IP of Cdc13 and Stn1 in a wild-type or an *elg1Δ* strain. Cells were cell-cycle synchronized in S-phase as the interaction between these two proteins was barely seen when immunoprecipitated in an asynchronous culture. When Elg1 was absent, Stn1 exhibited a strong reduction in its co-precipitation with Cdc13 (*Figure 5E and F*). This suggests that the interaction between Stn1 and Cdc13 is, at least partly, dependent on Elg1.

## Model: Elg1 negatively regulates the telomere length by forming an interaction with the CST complex

Taken together, our results suggest the following model (*Figure 6*).

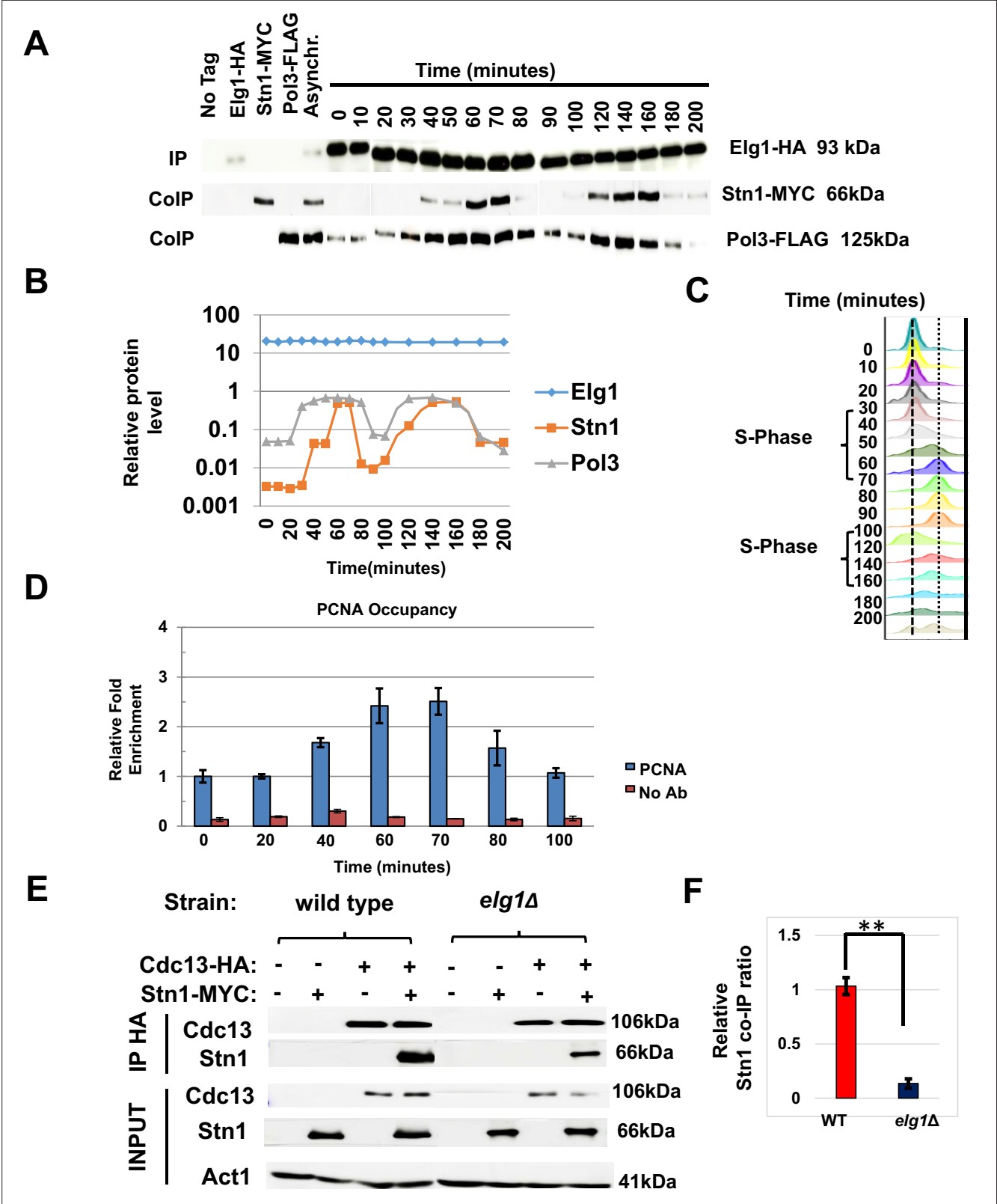

**Figure 5.** Timing of Elg1-Stn1 interaction. (**A**) Co-IP experiment with synchronized cells. Aliquots were taken at time intervals, Elg1 was immunoprecipitated, and the level of Stn1 and Pol3 (the large subunit of DNA polymerase Delta) was monitored by western blot. Strains with single tags are shown as controls. Whole-cell extract results are shown in *Figure 5—figure supplement 1*. (**B**) Quantitation of the western shown in (**A**). (**C**) DNA content of the cells used in (**A**) by cell cytometry. (**D**) Chromatin immunoprecipitation at telomeres (Telo-ChIP) in synchronized cells showing PCNA

*Figure 5 continued on next page*

*Figure 5 continued*

occupancy. (**E**) Interaction between Stn1 and Cdc13 in a wild type and a *elg1Δ* strain. (**F**) Quantitation of three independent biological repeats of the experiment shown in (**E**). **p<0.001.

The online version of this article includes the following source data and figure supplement(s) for figure 5:

**Source data 1.** The Source Data contains the original Western blots used to make the figure.

**Figure supplement 1.** Whole-cell extract showing the level of Elg1, Stn1, and Pol3 in the cell cycle experiment shown in *Figure 5*.

**Figure supplement 1—source data 1.** The Source Data contains the original Western blots used to generate the figure.

Following firing of the origin proximal to the telomeres at late S-phase, Elg1 moves with the fork, unloading PCNA from the lagging strand as the fork progresses (*Kubota et al., 2015*; *Kubota et al., 2013*; *Shemesh et al., 2017*; *Figure 6A*). SUMOylation of PCNA acts as a signal for the activation of telomerase (*Figures 1A, B and 6B*). Once Elg1 reaches the telomeres, it interacts with Stn1 (*Figure 5*). This interaction requires SUMOylation of Stn1 or of some other protein, which is not PCNA (*Figure 3D*) nor Cdc13 (*Figure 4—figure supplement 1A*) and could be Stn1 itself or another telomeric protein (*Hang et al., 2011*). The Elg1-Stn1 complex then unloads the SUMOylated PCNA at the telomeres, and Cdc13 becomes SUMOylated (*Figure 6C*), now interacting with Elg1-Stn1. SUMOylation of Cdc13 allows its interaction with Stn1, and presumably with Ten1 (*Hang et al., 2011*), creating the complete CST complex, and preventing further telomerase activity (*Figure 6D*). The fact that co-IP experiments detect a strong interaction with Stn1 but fail to detect an interaction with Cdc13 (*Figure 2B*) suggests that indeed the interaction Elg1-Cdc13 is very transient, and Elg1 leaves chromatin immediately, leaving the C-terminus of Stn1 free to interact with Cdc13. It is possible that

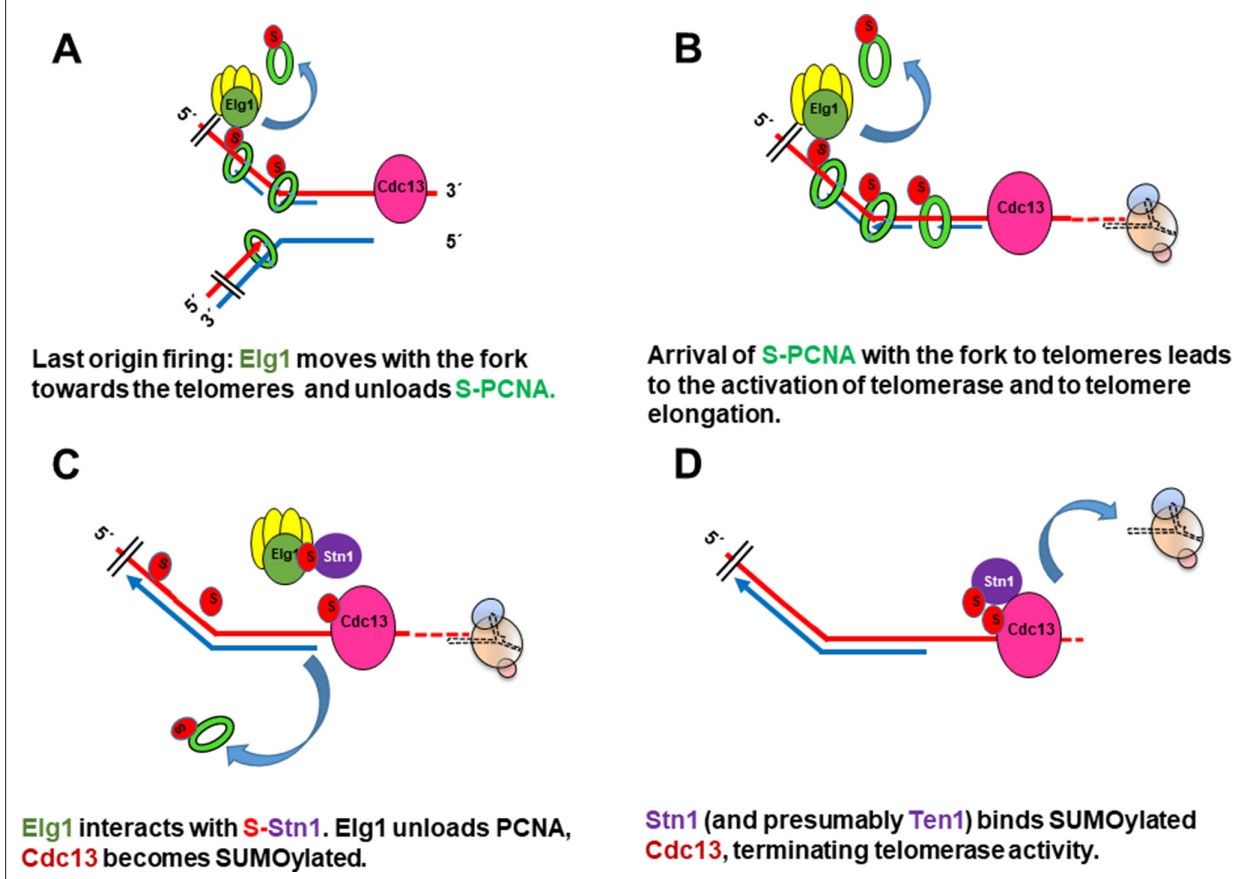

**Figure 6.** Model for the role of SUMOylated PCNA and Elg1 in telomere length regulation. (**A**) Cdc13 binds ssDNA at the telomeres, Elg1 moves with the replisome at the lagging strand, unloading PCNA in each Okazaki fragment. (**B**) Arrival of the SUMOylated PCNA at the fork to Cdc13 promotes telomerase activity. (**C**) Elg1 interacts with Stn1, which could be SUMOylated. Cdc13 becomes SUMOylated. Elg1 unloads PCNA and leaves telomeres. (**D**) Stn1 is now able to interact with Cdc13, evicting Est1 and terminating telomerase activity.

a single 'wave' of SUMOylation at the telomeres coordinates PCNA unloading with the interaction between Elg1 and Cdc13. Alternatively, unloading of PCNA by Elg1 may allow the recruitment of SUMO ligases to modify Cdc13. In support for this idea, YTH interactions between Stn1 or Cdc13 with SUMO are abolished in *siz1Δ siz2Δ* strains (*Figure 3D*).

## Discussion

We have presented evidence for the fact that two negative regulators of telomerase activity, Elg1 and Stn1, work in the same pathway to coordinate chromosomal DNA replication and telomere elongation. We present a model (*Figure 6*) in which the unloading of SUMOylated PCNA by Elg1 at telomeres facilitates the interaction between Stn1 and Cdc13 that negatively regulate telomerase activity.

This model is supported by several observations. First, the double mutants of Elg1 and Stn1 do not show increased telomere length phenotype in comparison to the single mutants, indicating that both proteins work in the same pathway that negatively regulates telomere length (*Figure 2A*). In addition, the interaction between Elg1 and Stn1 seems direct and occurs exclusively during the late S-phase (*Figures 2B and 5A–C*). This interaction is neither dependent on SUMOylation of PCNA (*Figure 3D*) nor on Cdc13 (*Figure 4—figure supplement 1A*). In contrast, the interaction between Cdc13 and Elg1 is dependent on both the SUMOylation of PCNA (*Figure 3D*) and of Cdc13 itself (*Figure 4F*). We suggest that Elg1 participates in the negative regulation of telomere length by unloading SUMOylated PCNA from the telomeres. This eliminates a positive signal for telomerase activity and causes the exchange of Est1 by Stn1 as interactor of Cdc13, effectively terminating telomerase activity (*Chandra et al., 2001*; *Gao et al., 2007*; *Ge et al., 2020*; *Hang et al., 2011*; *Puglisi et al., 2008*). As expected from the model, the interaction Elg1-Cdc13 is transient and cannot be detected by co-IP.

We present evidence for the fact that SUMOylated PCNA serves as a positive signal for telomere elongation (*Figure 1*). Preventing its unloading (by mutations in the *ELG1* gene) leads to elongated telomeres, and so does PCNA overexpression, provided the protein is able to be SUMOylated, or is fused to SUMO (thus mimicking SUMOylated PCNA) (*Figure 1*). *ELG1* is the only PCNA unloader in eukaryotes; however, it is not an essential gene, implying that there are alternative ways in which PCNA can get unloaded (either spontaneously or with the help of RFC) (*Arbel et al., 2021*). Once this step does take place, Stn1 can replace Est1 as Cdc13's partner and stop the activity of telomerase, but at a longer telomere steady-state equilibrium point. This explains both the long telomere phenotype of *elg1* mutants, and the fact that it is less severe than that of *stn1* alleles (*Figure 2*). The fact that *pol30-RR* mutants have stable, normal-sized telomeres (*Figure 1*) implies that despite lacking PCNA SUMOylation, they are able to solve the 'end replication problem.' Our results support the notion (*Maicher et al., 2017*) that there are two alternative modes of telomerase: one carrying 'basal activity,' with minimal genetic requirements, able to maintain most telomeres at normal size, and another, 'sustained activity,' required for increased elongating activity (*Maicher et al., 2017*). Previous results have shown that the first mode does not require Rnr1, the large subunit of the ribonucleotide reductase enzyme, whereas the second depends on it (*Maicher et al., 2017*). Similarly, SUMOylated PCNA is only needed for the second mode of telomerase. In our experiments, the phenotype that we monitored was elongated telomeres (e.g., by deletion of *ELG1* or by overexpression of PCNA) which required sustained telomerase activity. To solve the 'end replication problem' and maintain normal telomere length, very little telomerase activity might be needed, and even unmodified PCNA or telomeric proteins might be able to provide that basal level.

Many proteins in the cell become SUMOylated at one point or another. This modification is usually transient due to the activity of de-SUMOylases (which eliminate the modification) and STUbLs (SUMO-dependent ubiquitin ligases, which usually send the protein to degradation). It has been proposed that SUMOylation often works by targeting many physically interacting proteins together. This may be executed by the local recruitment of SUMO ligases to preassembled, co-localized protein groups. SUMOylation thus may increase their interactions, acting to increase their local concentration or effective interaction (*Jentsch and Psakhye, 2013*). SUMOylation of telomeric proteins seems to be one of those cases (*Garg et al., 2014*; *Hang et al., 2011*; *Matmati et al., 2018*; *Miyagawa et al., 2014*). In our model, PCNA, Cdc13, and possibly Stn1 undergo SUMOylation to ensure the proper sequence of events in the coordination between telomerase activity and chromosomal replication. Cdc13 binds Pol12, a subunit of Polymerase Alpha (*Grossi et al., 2004*; *Petreaca et al., 2006*; *Puglisi et al., 2008*), allowing the synthesis of the lagging strand on the newly synthesized ssDNA. Thus, the arrival of the

replisome to a short telomere at the end of S-phase brings PCNA, whose SUMOylation can elicit both telomerase activity, and its subsequent repression when PCNA is unloaded by Elg1. This in turn primes synthesis of the complementary strand, thus coordinating telomere elongation with chromosomal replication.

# Materials and methods

## Key resources table

| Reagent type (species) or resource | Designation | Source or reference | Identifiers | Additional information |
|---|---|---|---|---|
| Recombinant DNA reagent | pGBD424 | *Takahashi et al., 2020* | | Empty yeast two-hybrid vector |
| Recombinant DNA reagent | pGBD424-Pol30 | *Takahashi et al., 2020* | | Overexpresses wt PCNA |
| Recombinant DNA reagent | pGBD424-Pol30-RR | *Takahashi et al., 2020* | | Overexpresses unmodifiable PCNA |
| Recombinant DNA reagent | pGBD424-ubiquitin-Pol30-RR | *Takahashi et al., 2020* | | Overexpresses unmodifiable PCNA fused to ubiquitin |
| Recombinant DNA reagent | pGBD424-SUMO-Pol30-RR | *Takahashi et al., 2020* | | Overexpresses unmodifiable PCNA fused to SUMO |
| Recombinant DNA reagent | pGBU9 | *Parnas et al., 2010* | | Yeast two-hybrid vector |
| Strain, strain background (*Saccharomyces cerevisiae* MATa strain) | PJ69-4A | *James et al., 1996* | | Yeast two-hybrid strain |
| Chemical compound, drug | Alpha Factor | Sigma-Aldrich | T6901 | |
| Commercial assay or kit | Protein A sepharose beads | Sigma-Aldrich | 17-1279-01 | |
| Commercial assay or kit | Protein G sepharose beads | Sigma-Aldrich | 17-0618-01 | |
| Chemical compound, drug | Pronase | Sigma-Aldrich | P5147 | |
| Antibody | Anti-HA (mouse polyclonal) | Santa Cruz Biotechnology | sc7392 | 1:1000 |
| Antibody | Anti-Myc (mouse polyclonal) | Santa Cruz Biotechnology | 9E10, SC-40 | 1:1000 |

## Yeast strains, plasmids, and media

All yeast strains used are described in *Table 2*. All plasmids are described in *Table 3*. Yeast extract-peptone-dextrose (YPD) medium was prepared with a ready-to-use mixture (Formedium). Synthetic complete (SC) minimal medium was prepared with 2% dextrose (Formedium), yeast nitrogen base without amino acids (Difco), and all necessary amino acids. A 2% agar (Difco) was added for solid medium. Standard yeast genetic procedures were used to create single and double mutants. Unless stated otherwise, all experiments were carried out at 30°C.

## PCNA overexpression

The high copy number pGAD424 plasmid alone, or carrying the wild-type *POL30*, the mutant *pol30-KK127,164RR* (*pol30-RR*), or the *pol30-RR* fused to ubiquitin or to SUMO (*Parker et al., 2007*; *Takahashi et al., 2020*) were transformed into *pol30-RR* strains and grown under selective conditions.

## Southern teloblots

For each genotype, at least three independently created strains were tested after 10 consecutive passages on YPD plates (*Harari and Kupiec, 2018*; *Harari et al., 2013*; *Harari et al., 2017*; *Rubinstein et al., 2014*). Genomic DNA was isolated by pelleting 15 ml saturated overnight yeast culture grown in YPD, then beating the cells using 0.5 mm glass beads in 500 µl lysis buffer and 500 µl phenol chloroform. Cells were spun down for 10 min at 14,000 rpm, and ~500 µl of the top clear solution, containing the genomic DNA, was carefully taken out. DNA was precipitated using 1000 µl of 95% ethanol and then pelleted. The supernatant was discarded, and 500 µl of 70% ethanol was added. Pellets were left to dry before resuspension in 300 µl TE with RNaseA for 10 min at 65°C. The DNA was precipitated by adding 130 µl of 7.5 M ammonium acetate (pH 7) and 1 ml of 95% ethanol at −20°C,

**Table 2.** Yeast strains used in this study.

| Strain number | Name | Genotype |
|---|---|---|
| 13237 | BY4741 | MATa his3del, leu2del, met15del, ura3del |
| 12480 | BY elg1Δ::HygMX | MATa his3del, leu2del, met15del, ura3del, elg1::HYG |
| 14421 | BY pol30-K127R | MATa his3del, leu2del, met15del, ura3del, pol30-K127R::LEU2 |
| 14425 | BY pol30-K164R | MATa his3del, leu2del, met15del, ura3del, pol30-K164R::LEU2 |
| 14423 | BY pol30-KK127,164RR | MATa his3del, leu2del, met15del, ura3del, pol30-RR::LEU2 |
| 14426 | BY pol30-K127R elg1Δ | MATa his3del, leu2del, met15del, ura3del, pol30-K127R::LEU2, elg1::HYG |
| 14430 | BY pol30-K164R elg1Δ | MATa his3del, leu2del, met15del, ura3del, pol30-K164R::LEU2, elg1::HYG |
| 14428 | BY pol30-KK127,164RR elg1Δ | MATa his3del, leu2del, met15del, ura3del,pol30-RR::LEU2,elg1::HYG |
| 14398 | BY rad18Δ:: KanMX | MATa his3del, leu2del, met15del, ura3del, rad18::KanMX. |
| 14401 | BY rad18D::KanMX elg1D::HygMX | MATa his3del, leu2del, met15del, ura3del, rad18::KanMX, elg1::HYG |
| 19606 | BY pol30-KK127,164RR Elg1-HA | MATa his3del, leu2del, met15del, ura3del ELG1-HA-NAT pol30-KK127,164RR:HIS3 |
| 20622 | BY bar1 CDC13-HA ELG1-HA | MATa his3del, leu2del, met15del, ura3del, bar1::LEU2, CDC13:3HA:HISMX, ELG1-HA-KanMX |
| 18418 | BY STN1-Myc Elg1-HA | MATa his3del, leu2del, met15del, ura3del stn1::HYG CEN LEU2 STN1-(G)9-(myc)7 ELG1-HA-KanMx |
| 18790 | BY bar1 STN1-Myc ELG1-HA POL3-FLAG | MATa his3del, leu2del, met15del, ura3del bar1::NatMX stn1::HYG CEN LEU2 STN1-(G)9-(myc)7 ELG1-HA-KanMx POL3-FLAG-URA3 |
| 19552 | BY bar1 CDC13-HA POL30-FLAG | MATa his3del, leu2del, met15del ura3del, bar1:: NatMX CDC13-3HA::HISMX, POL30-FLAG::KanMX, |
| 20625 | BY CDC13-HA STN1-Myc | MATa his3del, leu2del, met15del, ura3del, CDC13-3HA::HISMX, stn1::HYG CEN LEU2 STN1-(G)9-(myc)7, |
| 20626 | BY CDC13-HA STN1-Myc elg1D | MATa his3del, leu2del, met15del, ura3del,CDC13-3HA::HISMX, stn1::HYG,CEN LEU2 STN1-(G)9-(myc)7,elg1::KanMx |
| 20623 | BY TEN1-FLAG | MATa his3del, leu2del, met15del, ura3del trp1del Lys2del can1:: STE2pr-Sp_HIS5 ten1::KanMX, Elg1-HA-NAT CEN URA3 TEN1-(G)8-(FLAG)3 |
| 17611 | W303-1a | MATa leu2-3, 112 ura3-1 his3-11,15, trp1-1, ade2-1, can1- 100 |
| 9842 | W303 elg1Δ::HygMX | MATa leu2-3, 112 ura3-1 his3-11,15, trp1-1, ade2-1, can1- 100, elg1::KanMx |
| 9551 | W303 stn1-13 | MATa leu2-3, 112 ura3-1 his3-11,15, trp1-1, ade2-1, can1- 100, stn1-13 |
| 9848 | W303 elg1Δ::HygMX stn1-13 | MATa leu2-3, 112 ura3-1 his3-11,15, trp1-1, ade2-1, can1- 100 elg1::KanMx, stn1-13 |
| 12357 | W303 stn1-164 | MATa leu2-3, 112 ura3-1 his3-11,15, trp1-1, ade2-1, can1- 100, stn1Δ::Hyg+pRS-stn1-L164A (pVL3571) |
| 12358 | W303 stn1-164 elg1Δ: KanMX | MATa leu2-3, 112 ura3-1 his3-11,15, trp1-1, ade2-1, can1- 100, elg1::KanMx, stn1Δ::Hyg+pRS-stn1-L164A (pVL3571) |
| 12062 | PJ69-4 | MAT@ trp1-901 leu2-3,112 ura3-52 his3-200 gal4del gal80del GAL2-ADE2 LYS2:: GAL1-HIS3 met2::GAL7-lacZ |
| 15017 | PJ elg1D | MAT@ trp1-901 leu2-3,112 ura3-52 his3-200 gal4del gal80del GAL2-ADE2 LYS2:: GAL1-HIS3 met2::GAL7-lacZ elg1::HYG |
| 19774 | PJ siz1 siz2 | MAT@ trp1-901 leu2-3,112 ura3-52 his3-200 gal4del gal80del GAL2-ADE2 LYS2:: GAL1-HIS3 met2::GAL7-lacZ siz1::KanMX siz2::HYG |
| 11069 | PJ pol30-RR | MAT@ trp1-901 leu2-3,112 ura3-52 his3-200 gal4del gal80del GAL2-ADE2 LYS2:: GAL1-HIS3 met2::GAL7-lacZ POL30-RR:: leu2:: KANMX |
| 20624 | PJ STN1-Myc | MAT@ trp1-901 leu2-3,112 ura3-52 his3-200 gal4del gal80del GAL2-ADE2 LYS2:: GAL1-HIS3 met2::GAL7-lacZ elg1-sim::KanMx, stn1::HYG,CENLEU2 STN1-(G)9-(myc)7 |
| 19916 | PJ elg1-sim-Myc | MAT@ trp1-901 leu2-3,112 ura3-52 his3-200 gal4del gal80del GAL2-ADE2 LYS2:: GAL1-HIS3 met2::GAL7-lacZ elg1-SIM-MYC-KANMX |
| 18798 | PJ elg1-DDMyc | MAT@ trp1-901 leu2-3,112 ura3-52 his3-200 gal4del gal80del GAL2-ADE2 LYS2:: GAL1-HIS3 met2::GAL7-lacZ elg1-DD-MYC-KANMX |

*Table 2 continued on next page*

*Table 2 continued*

| Strain number | Name | Genotype |
|---|---|---|
| 19917 | PJ elg1-DD+sim-Myc | *MAT@ trp1-901 leu2-3,112 ura3-52 his3-200 gal4del gal80del GAL2-ADE2 LYS2:: GAL1-HIS3 met2::GAL7-lacZ elg1-DDsim-MYC-KANMX* |
| 19915 | PJ elg1-Walker A | *MAT@ trp1-901 leu2-3,112 ura3-52 his3-200 gal4del gal80del GAL2-ADE2 LYS2:: GAL1-HIS3 met2::GAL7-lacZ, elg1 KK343/4DD:MYC:KanMX* |
| 19486 | PJ cdc13-snm | *MAT@ trp1-901 leu2-3,112 ura3-52 his3-200 gal4del gal80del GAL2-ADE2 LYS2:: GAL1-HIS3 met2::GAL7-lacZ cdc13-snm* |

spun, and resuspended in 50 µl of TE. About 10 µl of genomic DNA was digested using XhoI overnight at 37°C. Digested gDNA and ladder were loaded onto a 0.8% agarose gel and electrophoresed overnight at ~67 V in 1× TBE. The gel was denatured for 30 min in a rocking shaker (1.5 M NaCl, 0.5 M NaOH) and was neutralized for 15 min (1.5 M NaCl, 0.5 M Tris, pH 7.0). The DNA on the gel was transferred onto a hybond nylon membrane (GE Healthcare GERPN303B) with 10× SSC (1.5 M NaCl 0.17 M NaCitrate, dihydrate) and UV cross-linked before blocking in Church buffer (0.5 M $Na_2HPO_4$, pH 7.2, 7% SDS, 1 mM EDTA, 1% BSA) for ~2 hr at 65°C. $^{32}$P radiolabeled PCR fragments were added onto the membrane and left to incubate overnight. The membrane was washed 3–4 times for 15 min each, with 1× SSC 0.1%SDS buffer before drying and exposing it to X-ray film. The teloblots blots were exposed for 1–8 d at –70°C before the film was developed.

**Table 3.** Plasmids used in this study.

| Strain number | Genotype | Source or reference |
|---|---|---|
| 4239 | pGad424 (LEU2) | Kupiec lab |
| 4238 | pGad424-POL30 | Helle Ulrich lab |
| 4237 | pGad424-pol30-k127r,k164r | Helle Ulrich lab |
| 4236 | pGad424-pol30-k127r,k164r-SMT3 | Helle Ulrich lab |
| 4235 | pGAD424-pol30-k127r,k164r-UBI | Helle Ulrich lab |
| 4147 | CEN LEU2 STN1-(G)9-(myc)7 | Victoria Lundblad lab |
| 4144 | CEN URA3 TEN1-(G)8-(FLAG)3 | Victoria Lundblad lab |
| 2201 | pCN181 pACT2-STN1 (LEU2) | Constance Nugent lab |
| 2205 | pVL855 pACT2-CDC13 | Constance Nugent lab |
| 2168 | pACT2 - TEN1 | Michel Charbonneau lab |
| 1775 | pGBU9 (URA3) | Kupiec lab |
| 1973 | pGBU9-ELG1-NTD(1-234) | This study |
| 3301 | pGBU9-ELG1-CTD(541-791) | This study |
| 2260 | pGBU9-ELG1-NTDsim(1-234) | This study |
| 2241 | pGBU9-SMT3 | This study |
| 2419 | pGAD424-Stn1-Nt(1-282) | David Shore lab |
| 2420 | pGAD424-Stn1-Ct(282-494) | David Shore lab |
| 2169 | pGAD424-Stn1-13 | Michel Charbonneau lab |
| 2418 | pGAD424-Stn1-63 | David Shore lab |
| 4287 | pACT-cdc13-snm | This study |

## Telomeric probe

*S. cerevisiae*-specific telomeric probe was labeled by random priming of 20–25 ng of a telomeric fragment and of 20–25 ng of size-control fragments using the DNA labeling mix from Biological Industries. Both fragments were generated by PCR using the primers indicated below. For the telomeric probe, a specific region of the left telomere of chromosome VII was amplified to generate a product 370 bp long. As size-control probe, a specific region of chromosome II was amplified to generate a product 1100 bp long. Since it contains an XhoI site, the size-control probe detects two bands in the sizes of 2044 bp, 779 bp in the Southern blot.

The primers for the Y' element product are

Y' element forward: GTTGGAGTTTTTCAGCGTTTGC
Y' element reverse: TGTGAACCGCTACCATCAGC
The Y' element PCR product is ~370 bp
The primers for the control product are:
Tel control forward: TTGTAGGGGCCTTTTGTAATGT
Tel control reverse: GTGCGCCCAGTAAGGGGT
The control PCR product is ~1100 bp

Telomere length was determined using TelQuant (*Rubinstein et al., 2014*).

## Yeast two-hybrid

The yeast two-hybrid assays were performed using PJ69-4A (*James et al., 1996*) strains cotransformed with a *LEU2*-marked plasmid containing genes fused to the *GAL4* activating domain (pACT or pGAD424) and a *URA3*- marked plasmid containing genes fused to the *GAL4* DNA binding domain (pGBU9). Strains containing the test plasmids were grown for 24 hr at 30°C in SC-Ura-Leu liquid medium and were plated on the reporter maker SC-His medium.

## Protein extraction and immunoprecipitation (IP) assays

Cells were grown to mid-logarithmic phase, washed once with water, and resuspended in lysis buffer (20 mM Tris-HCl, pH 7.5, 0.5 mM EGTA, 0.5 mM EDTA, 1 mM DTT, 125 mM potassium acetate, 12.5% glycerol, 0.1% Triton X-100, protease inhibitor mixture, and 1 mM phenylmethylsulfonyl fluoride). Cells were broken for 45 min with glass beads, centrifuged for 10 min at 10,000 × *g*, and the supernatant was collected. 20–30 µg of total protein extract was resolved on SDS-PAGE using 10% acrylamide gels. For immunoprecipitation, 1000 µg of proteins were prepared and pre-cleared with 20 µl of protein A-Sepharose and protein G-Sepharose beads mixture (GE Healthcare). 2 µl of (HA, Santa Cruz Biotechnology [sc7392; 1:1000] or MYC, Santa Cruz Biotechnology [9E10, SC-40; 1:1000]) antibodies were added to the cleared extract and incubated overnight at 4°C. The beads were washed once with lysis buffer, once with lysis buffer containing 0.5 m NaCl, and twice with buffer A (50 mM Tris-HCl, pH 7.5, 0.1 mM EGTA, 0.1% β-mercaptoethanol). The resulting immunoprecipitates were used for in vitro kinase assays.

## Western blotting

Cells were collected by centrifugation, resuspended in 600 ul of phosphate-buffered saline with 1% Triton X-100 (PBST), supplemented with a protease inhibitor cocktail (Roche), and subjected to mechanical rupture using glass beads. The cell debris were removed by centrifugation, and the supernatants were applied onto 0.1 M dithiothreitol, and incubated at 80°C for 10 min before sodium dodecyl sulfate-polyacrylamide gel electrophoresis (SDS-PAGE) (resolving gel: 30% acrylamide, 1.5 M Tris-HCl pH 8.8, 10% SDS [pH 7.2], 9.7 ml $H_2O$, 100 ul 10% APS, and 10 ul TEMED; stacking gel: 30% bis/acrylamide, 1 M Tris-HCl pH 6.8, 10% SDS [pH 7.2], 5.5 ml $H_2O$, 800 ul 10% APS, and 8 ul TEMED). The samples were run with SDS-PAGE buffer at 100 V until the samples have passed the stacking gel and then at 160 V until the samples have been fully separated. Transfer to nitrocellulose was done in transfer buffer (200 ml methanol, 3.03 g Tris base, 14.4 g glycine) at 500 mAmp and verified by staining with Ponceau-S dye. The blot was blocked with Milk for at least 60 min at room temperature. Primary antibody was added for 12 hr at 4°C. The blot was washed 3 × 5 min with TBST (Tris-buffered saline Tween-20) and secondary antibody was added for 1 hr. The blot was washed 3 × 5 min with TBST and subjected to electro-chemiluminiscence.

## Cell cycle synchronization

To assay Elg1 and Stn1 interaction during the cell cycle, the triple-tagged strain was grown in YPD to $OD_{600}$ 0.6–0.8, followed by the addition of 500 ng/ml α factor and was grown until ~90% of the cells appeared unbudded or exhibit Shmoo formation (for ~2 hr). The α-factor was removed by centrifugation and washing cells 2–3 times with warm YPD. Cells were released into YPD at $OD_{600}$ 0.6–0.8 with addition of pronase at a final concentration of 0.1 mg/ml. Cells were collected for both FACS and western blot analysis at different time points after release. Cells for FACS analysis were fixed in 70% ethanol, digested with RNase overnight, washed again and stained with propidium iodide (15 μg/ml), and analyzed by flow cytometry.

## Flow cytometry

200 μl of a logarithmic cell culture (0.6 $OD_{600}$) were harvested, resuspended in 60 μl of 50 mM Tris pH 7.5, and 140 μl of ethanol was added; cells were then kept overnight at 4°C. Fixed cells were centrifuged and washed once in 200 μl of 50 mM Tris pH 7.5 buffer and resuspended in 100 μl RNAse (0.2 mg/ml in 50 mM Tris pH 7.5) for 2 hr at 37°C. In addition, proteinase-K (0.2 mg/ml in 50 mM Tris pH 7.5) was added to each tube and cells were incubated for 60 additional minutes at 50°C. 20 μl of the sample was taken into a new tube and a 180 μl of 18 μg/ml propidium iodide 50 mM Tris pH 7.5 was added. The samples were kept in the dark at 4°C overnight, sonicated twice at low setting (20% power) for 3–5 s, and stored in the dark at 4°C. The flow cytometry MACSQuant system was used for reading. Results were analyzed using either the Flowing Software or the FlowJo program.

## Chromatin immunoprecipitation

50 ml of each strain were grown to $OD_{600} \approx 1$ in YPD. 1.5 ml formaldehyde (37% solution) was added for 15 min, and the formaldehyde was quenched with 2.5 ml of 2.5 M glycine for 5 min. Cells were harvested, washed once with 15 ml cold PBS, and broken down for 10 min with glass beads in 600 μl lysis buffer (50 mM HEPES-KOH pH 7.5, 140 mM NaCl, 1 mM EDTA, 1% Triton X100, 0.1% Na-deoxycholic acid). The supernatant (lysate) was removed to a new tube. The glass beads were washed with 500 μl lysis buffer, centrifuged, and the supernatant was added to the lysate. The lysate was sonicated 6–8 times for 10–15 s at 80% amplitude with 1 min on ice between each time. The sonicated material was centrifuged for 20 min at 2500 rpm. The supernatant was used for immunoprecipitations (IP). The sonicated proteins were pre-cleared with a 25 μl protein A sepharose and protein G sepharose beads mixture (GE Healthcare), and the appropriate antibodies were added to the cleared extract and incubated overnight at 4°C. PCNA were immunoprecipitated with 2–5 μg of anti-PCNA antibody (Sigma). A total of 10% of the extract was saved as input. The beads after the IP were washed once with lysis buffer, once with lysis buffer with 360 mM NaCl, once with washing buffer (10 mM Tris/HCl pH 8, 0.25 M LiCl, 0.5% NP40, 0.5% Na-deoxycholic acid, 1 mM EDTA), and once with TE (10 mM Tris/HCl pH 8 and 10 mM EDTA). The washed beads and the input were treated with elution buffer (50 mM Tris/HCl pH 8, 10 mM EDTA, 1% SDS) overnight at 65°C. The DNA was precipitated, resuspended in water, and used for PCR real-time analysis (ABI StepOnePlus Real-Time PCR System); primer concentration and cycles number were calibrated individually for each reaction. All experiments are plotted as the average of at least three independent biological repeats, and each biological repeat is the average of three technical PCR repeats. The oligonucleotides used are

Y'-element 5'-GGCTTGATTTGGCAAACGTT-3'
5'-GTGAACCGCTACCATCAGCAT-3'

## Materials availability

All materials in this article can be available upon contact with the corresponding author.

## Acknowledgements

We thank David Shore, Hele Ulrich, Connie Nugent, Michel Charboneau, and Vicky Lundblad for strains and plasmids. We thank all present and past members of the Kupiec lab for support and ideas, and Ofir Hurvitz for help during the first stages of this project. This research was supported by grants from the Israel Science Foundation, the Israel Cancer Research Fund, and the Recanati Fund.

## Additional information

### Funding

| Funder | Grant reference number | Author |
|---|---|---|
| Israel Science Foundation | 1105/19 | Martin Kupiec |
| Israel Cancer Research Fund | 408 | Martin Kupiec |
| Recanati Fund | 23 | Martin Kupiec |

The funders had no role in study design, data collection and interpretation, or the decision to submit the work for publication.

### Author contributions

Pragyan Singh, Conceptualization, Formal analysis, Investigation, Writing – original draft; Inbal Gazy, Conceptualization, Formal analysis, Investigation; Martin Kupiec, Conceptualization, Resources, Data curation, Formal analysis, Supervision, Funding acquisition, Investigation, Writing – original draft, Project administration, Writing – review and editing

### Author ORCIDs

Martin Kupiec ⬚ https://orcid.org/0000-0002-7934-3342

Reviewer #1 (Public Review): https://doi.org/10.7554/eLife.86990.3.sa1
Reviewer #2 (Public Review): https://doi.org/10.7554/eLife.86990.3.sa2
Reviewer #3 (Public Review): https://doi.org/10.7554/eLife.86990.3.sa3
Author Response https://doi.org/10.7554/eLife.86990.3.sa4

## Additional files

### Supplementary files

• MDAR checklist

### Data availability

All data generated or analysed during this study are included in the manuscript and supporting file; Source Data files have been provided for Figures 1A, 1B, 2A, 2B, 3E, 4-Suppl. 1, 5A, 5E, 5-Suppl. 1.

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
