## [Editor Report · eLife assessment]

This **important** study aims to discover the mechanisms governing the switch between conventional DNA replication and the specialized mechanism of telomere end replication. **Solid** genetic and biochemical assays suggest an interplay between sumoylated PCNA and chromosome terminal capping proteins. The questions addressed have implications for several fields, such as genome stability.

---

## [Referee Report · Reviewer #1 (Public Review)]

In this manuscript, the authors investigated the role of Elg1 in the regulation of telomere length. The main role of the Elg1/RLC complex is to unload the processivity factor PCNA, mainly after completion of synthesis of the Okazaki fragment in the lagging strand. They found that Elg1 physically interacts with the CST (Cdc13-Stn1-Ten1) and propose that Elg1 negatively regulates telomere length by mediating the interaction between Cdc13 and Stn1 in a pathway involving SUMOylation of both PCNA and Cdc13. Accumulation of SUMOylated PCNA upon deletion of ELG1 or overexpression of RAD30 leads to elongated telomeres. On the other hand, the interaction of Elg1 with Sten1 is SIM-dependent and occurs concurrently with telomere replication in late S phase. In contrast Elg1-Cdc13 interaction is mediated by PCNA-SUMO, is independent on the SIM of Elg1 but still dependent on Cdc13 SUMOylation. The authors present a model containing two main messages (1) PCNA-SUMO acts as a positive signal for telomerase activation (2) Elg1 promotes Cdc13/Stn1 interaction at the expense of Cdc13/Est1 interaction thus terminating telomerase action.

The manuscript contains a large amount of data that make a major inroads on a new type of link between telomere replication and regulation of the telomerase. Nevertheless, the detailed choreography of the events as well as the role of PCNA-SUMO remain elusive and the data do not fully explain the role of the Stn1/Elg1 interaction. The data presented do not convincingly support the claim that SUMO-PCNA is a positive signal for telomerase activation. This was partially addressed in the current version.

---

## [Referee Report · Reviewer #2 (Public Review)]

This paper purports to unveil a mechanism controlling telomere length through SUMO modifications controlling interactions between PCNA unloader Elg1 and the CST complex that functions at telomeres. This is an extremely interesting mechanism to understand, and this paper indeed reveals some interesting genetic results, leading to a compelling model, with potential impact on the field. Overall, however, the data do not provide sufficient support for the claims. The model may be correct but it is not yet convincingly demonstrated.

The current version addressed some of the issues regarding language describing conclusions and more experimental detail has been provided. However, the authors have not provided new data supporting the model, so the overall evaluation is that the work remains inconclusive.

---

## [Referee Report · Reviewer #3 (Public Review)]

This paper reveals interesting physical connections between Elg1 and CST proteins that suggest a model where Elg1-mediated PCNA unloading is linked to regulation of telomere length extension via Stn1, Cdc13, and presumably Ten1 proteins. Some of these interactions appear to be modulated by sumolyation and connected with Elg1's PCNA unloading activity. The strength of the paper is in the observations of new interactions between CST, Elg1, and PCNA. These interactions should be of interest to a broad audience interested in telomeres and DNA replication.

What is not well demonstrated from the paper is the functional significance of the interactions described. The model presented by the authors is one interpretation of the data shown, and proposes that the role of sumolyation is temporally regulate the Elg1, PCNA and CST interactions at telomeres. This model makes some assumptions that are not demonstrated by this work (such as Stn1 sumolyation, as noted) and are left for future testing. Alternative models that envision sumolyation as a key in promoting spatial localization could also be proposed based on the data here (as mentioned in the discussion), in addition to or instead of a role for sumolyation in enforcing a series of switches governing a tightly sequenced series of interactions and events at telomeres. Critically, the telomere length data from the paper indicates that the proposed model depicts interactions that are not necessary for telomerase activation or inhibition, as telomeres in pol30-RR strains are normal length and telomeres in elg1∆ strains are not nearly as elongated as in stn1 strains. One possibility mentioned in the paper is the PCNAS and Elg1 interactions are contributing to the negative regulation of telomerase under certain conditions that are not defined in this work. Could it also be possible that the role of these interactions is not primarily directed toward modulating telomerase activity? It will be of interest to learn more about how these interactions and regulation by Sumo function intersect with regulation of telomere extension.

---

## [Author Response]

The following is the authors’ response to the original reviews.

**Reviewer #1 (Public Review):**
In this manuscript, the authors investigated the role of Elg1 in the regulation of telomere length. The main role of the Elg1/RLC complex is to unload the processivity factor PCNA, mainly after completion of synthesis of the Okazaki fragment in the lagging strand. They found that Elg1 physically interacts with the CST (Cdc13-Stn1- Ten1) and propose that Elg1 negatively regulates telomere length by mediating the interaction between Cdc13 and Stn1 in a pathway involving SUMOylation of both PCNA and Cdc13. Accumulation of SUMOylated PCNA upon deletion of ELG1 or overexpression of RAD30 leads to elongated telomeres. On the other hand, the interaction of Elg1 with Sten1 is SIM-dependent and occurs concurrently with telomere replication in late S phase. In contrast Elg1-Cdc13 interaction is mediated by PCNA-SUMO, is independent on the SIM of Elg1 but still dependent on Cdc13 SUMOylation. The authors present a model containing two main messages (1) PCNA- SUMO acts as a positive signal for telomerase activation (2) Elg1 promotes Cdc13/Stn1 interaction at the expense of Cdc13/Est1 interaction thus terminating telomerase action.The manuscript contains a large amount of data that make a major inroad on a new type of link between telomere replication and regulation of the telomerase.Nevertheless, the detailed choreography of the events as well as the role of PCNA- SUMO remain elusive and the data do not fully explain the role of the Stn1/Elg1 interaction. The data presented do not sufficiently support the claim that SUMO- PCNA is a positive signal for telomerase activation.

We thank the reviewer for her/his review efforts and opinion. We have re-submitted a new version of the manuscript in which we clarify some of the criticisms presented. In a point-by-point letter we respond to all the specific queries.

**Reviewer #2 (Public Review):**
This paper purports to unveil a mechanism controlling telomere length through SUMO modifications controlling interactions between PCNA unloader Elg1 and the CST complex that functions at telomeres. This is an extremely interesting mechanism to understand, and this paper indeed reveals some interesting genetic results, leading to a compelling model, with potential impact on the field. The conclusions are largely supported by experiments examining protein-protein interactions at low resolution and ambiguous regarding directness of interactions like co-IP and yeast two-hybrid (Y2H) combined with genetics. However, some results appear contradictory and there's a lack of rigor in the experimental data needed to support claims. There is significant room for improvement and this work could certainly attain the quality needed to support the claims. The current version needs substantial revision and lacks the necessary experimental detail. Stronger support for the claims would add detail to help distinguish competing models.

We thank the reviewer for her/his positive opinion. We have re-submitted a new version of the manuscript in which we clarify some of the criticisms presented by thereferees, and added all the missing experimental details. In a point-by-point letter we respond to all the specific queries.

**Reviewer #3 (Public Review):**
This paper reveals interesting physical connections between Elg1 and CST proteins that suggest a model where Elg1-mediated PCNA unloading is linked to regulation of telomere length extension via Stn1, Cdc13, and presumably Ten1 proteins. Some of these interactions appear to be modulated by sumolyation and connected with Elg1's PCNA unloading activity. The strength of the paper is in the observations of new interactions between CST, Elg1, and PCNA. These interactions should be of interest to a broad audience interested in telomeres and DNA replication.

We thank the reviewer for her/his positive opinion. We have re-submitted a new version of the manuscript in which we clarify some of the criticisms presented. In a point-by-point letter we respond to all the specific queries.

What is not well demonstrated from the paper is the functional significance of the interactions described. The model presented by the authors is one interpretation of the data shown, and proposes that the role of sumolyation is temporally regulate the Elg1, PCNA and CST interactions at telomeres. This model makes some assumptions that are not demonstrated by this work (such as Stn1 sumolyation, as noted) and are left for future testing. Alternative models that envision sumolyation as a key in promoting spatial localization could also be proposed based on the data here (as mentioned in the discussion), in addition to or instead of a role for sumolyation in enforcing a series of switches governing a tightly sequenced series of interactions and events at telomeres. Critically, the telomere length data from the paper indicates that the proposed model depicts interactions that are not necessary for telomerase activation or inhibition, as telomeres in pol30-RR strains are normal length and telomeres in elg1∆ strains are not nearly as elongated as in stn1 strains. One possibility mentioned in the paper is the PCNAS and Elg1 interactions are contributing to the negative regulation of telomerase under certain conditions that are not defined in this work. Could it also be possible that the role of these interactions is not primarily directed toward modulating telomerase activity? It will be of interest to learn more about how these interactions and regulation by Sumo function intersect with regulation of telomere extension.

We present compelling evidence for a role of SUMOylated PCNA in telomere length regulation. Figure 1 shows that this modification is both necessary and sufficient to elongate the telomeres, indicating that PCNA SUMOylation plays a positive role in telomere elongation. The model we present is consistent with all our results. There are, of course, possible alternative models, but they usually fail to explain some of the results. We agree that the fact that pol30-RR presents normal-sized telomeres implies that SUMO-PCNA is not required for telomerase to solve the "end replication problem", but rather is needed for "sustained" activity of telomerase. Since elongated telomeres (by absence of Elg1 or by over-expression of SUMO-PCNA) was the phenotype monitored, this may require sustained telomerase activity. Similar results were seen in the past for Rnr1 (Maicher et al., 2017), and this mode depends on Mec1, rather than Tel1 (Harari and Kupiec, 2018). Telomere length regulation is complex, and we may not yet understand the whole picture. It appears that for normal “end replication problem” solution, very little telomerase activity may be needed, and spontaneous interactions at a low level may suffice. Future work may find the conditions at which telomerase switches from "end replication problem" to "sustained" activity. We have added further explanations on this subject to the Discussion section.

We suspect, but could not prove, a role for Stn1 SUMOylation in the interactions. SUMOylation is usually transient, and notoriously hard to detect, and despite the fact that many telomeric proteins are SUMOylated, Stn1 SUMOylation could not be shown directly by us and others (Hang et al, 2011).

**Reviewer #1 (Recommendations For The Authors):**
Suggestions for improved or additional experiments, data or analyses.My main concern is the claim that SUMOylated PCNA acts as a positive signal for telomerase activation. Yet the pol30-RR mutant has no impact on telomere length. The explanation of the authors is not entirely convincing.

We are aware that the regulation of telomere length is complex, and we may not fully understand it yet. Just consider the fact that ~500 genes participate in determining the final telomere length of a yeast (Askree et al., 2004). Since mutation in EACH of these genes has a phenotype, the implication is that the joint action of 500 players determines the outcome (a dialogue of 500 participants). Having said this, we clearly show in figure 1 that mutations that prevent PCNA SUMOylation prevent telomere length elongation in cells lacking Elg1, and overexpressing SUMOylated PCNA is enough to elongate the telomeres. Thus, SUMOylation of PCNA does act as a positive signal for elongation.

However, it appears that to fulfill the minimal requirement of dealing with the "end- replication problem", PCNA SUMOylation is not required, and only a "sustained activity" mode requires the S-PCNA signal (as we have also shown, surprisingly, for RNR1, Maicher et al. 2017). This sustained activity mode depends on Mec1, rather than Tel1 (Harari and Kupiec, 2018). Since elongated telomeres (by absence of Elg1 or by over-expression of SUMO-PCNA) was the phenotype monitored, this may require sustained telomerase activity. Telomere length regulation is complex, and we may not yet understand the whole picture. It appears that for normal “end replication problem” solution, very little telomerase activity may be needed, and spontaneous interactions at a low level may suffice for example, unmodified PCNA may promote telomerase activity at a lower level than that of SUMO-PCNA. Future work may find the conditions at which telomerase switches from "end replication problem" to "sustained" activity.

We have added further explanations on this subject to the Discussion section.

The model is entitled « Elg1 negatively regulates the telomere length by forming an interaction with the CST complex ». Nevertheless, expression of PCNA-RR completely reversed the long telomere phenotype of elg1∆ cells. Thus it appears that although the interaction between Stn1 and Cdc13 is reduced in the absence of Elg1, Elg1/Stn1 interaction is not instrumental in the formation of the CST complex and thus in the termination of telomerase activity. Does the elg1∆SIM mutant that does not interact with Stn1 impact telomere length?In the model part (lane 318), it is argued that the complex Elg1-Stn1 unloads SUMOylated PCNA. Elg1-Stn1 interaction depends on the SIM of Elg1. This SIM is however not required for Elg1's function in genome-wide SUMO-PCNA unloading, is it required specifically at telomeres?

The interactions between Elg1 and SUMOylated PCNA are carried out through both the SIM and the Threonines 386 and 387 (Shemesh et al, 2017). Consistently, the single elg1-SIM mutant has telomeres of normal length, and its effects on telomere length can only be seen when combined with mutations in the Threonines (elg1- TT386/7AA or elg1-TT386/7DD). Although the unloading of SUMOylated PCNA by Elg1 is important, the gene is not essential, and PCNA is either eventually unloaded by RFC, or spontaneously dis-assembles. This explains why the telomere length does not reach the same length in the absence of Elg1 as in the absence of, say, Stn1.

The model suggests that Elg1 promotes the interaction between Cdc13 and Stn1. This is based on the data presented in Figure 5 E and F. This is an important result. Because the experiment has been done on cells synchronized in S phase and the Elg1/Stn1 interaction occurs specifically at the end of S-phase, the FACS profile should be shown or a control provided to show that the two conditions are comparable.

The FACS profile for this experiment is shown in Figure 5C.

Does the interaction between Cdc13 and Pol30 depend on the SUMOyaltion of POL30 ?

Yes. We have added this as new Figure S2, and presented the results together with Figure 3 (Figure 3 is already too crowded).

Others points :Fig 1 : it should be mentioned in the Materials and Methods or in the figure legend how the average telomere lengths (horizontal bar) were calculated from the teloblot, as the position of the bar is not always intuitive

We estimate telomere length by using TelQuant (Rubinstein et al., 2014). We have added this to the Methods section.

-Fig 2 : Owing to the large span of telomere length in the stn1 mutants, the epistatic relationship between elg1∆ and stn1 mutants is poorly illustrated by the teloblot.

We repeated this experiment several times, and stn1 mutants consistently gave a very spread telomere length. In ALL the blots, however, the double mutants elg1 stn1 showed a telomere length similar to that of the single stn1 mutant, and never longer.

It is mentioned that other mutants in the collection showed epistasis. Are any of these mutants related to telomere replication or the proposed model?

Since we used the collection of non-essential mutants (so far), it was quite devoid of genes involved in DNA replication, which are mostly essential. An exception was siz1Δ, which showed epistasis with elg1Δ.

The section entitled « Elg1's functional activity is essential for its interaction with Cdc13 » (lane 205) is difficult to follow. The hierarchy between the different mutants of Elg1 on their capacity to unload PCNA is not totally in agreement with the data published in Itzkovich et al 2023 and Shemesh et al. 2017. In particular it appears to me from these papers that elg1-WalkerA 238 (KK343/4AA) mutant did not show a defect in contrast to elg1-WalkerA 238(KK343/4DD).

We are sorry for the typo in the results. We used the elg1-WalkerA (KK343/4DD) allele, which has a normal SIM but no activity. In a nutshell, we used mutants that either did or did not show unloading activity and/or SIM. The results clearly show that you need to unload PCNA in order for the N-ter of Elg1 to interact with Cdc13.

Are the synchronization done at 30{degree sign}C ?

Yes. We have added the information to the Methods section.

ChIP experiments are not described in the Materials and Methods

We apologize for this. They are now described.

In the figure 6, the PCNA rings are curiously placed at the beginning of the Okasaki fragments.

We thank the referee for noticing, we have corrected the figure.

**Reviewer #2 (Recommendations For The Authors):**
This paper purports to unveil a mechanism controlling telomere length through SUMO modifications controlling interactions between PCNA unloader Elg1 and the CST complex that functions at telomeres. This is an extremely interesting mechanism to understand, and this paper indeed reveals some interesting genetic results, leading to a compelling model, with potential impact on the field. The conclusions are largely supported by experiments examining protein-protein interactions at low resolution and ambiguous regarding directness of interactions like co-IP and yeast two-hybrid (Y2H) combined with genetics. However, some results appear contradictory and there's a lack of rigor in the experimental data needed to support claims. There is significant room for improvement and this work could certainly attain the quality needed to support the claims. The current version needs substantial revision and lacks necessary experimental detail. Stronger support for the claims would add detail to help distinguish competing models.Specific comments:Insufficient technical detail: I could find no explanation of how overexpression was achieved. No description of how teloChIP is performed, either for the PCNA IP or how the sequence analysis is performed. Too limited details on growth like exact temperatures for the cell cycle time course.

We have significantly expanded the Methods section to include all the technical information.

Please do not bold and underline text for emphasis-EVER

We have removed those from the text.

Lines 130-132: they have not shown "accumulation of SUMOylated PCNA" anywhere; this is an inference.

We have modified the text, it says: ”show that SUMOylated PCNA, and not unmodified or ubiquitinated PCNA, is both necessary and sufficient for telomere elongation in the presence or in the absence of Elg1.”

Fig 2A Can authors show any other very long-telomere mutant like stn1 that does show enhancement in combination with elg1∆ to show feasibility of such phenotype?

We don't think it is appropriate for the paper, but we have systematically created double mutants with elg1Δ and found many additive and even synergistic interactions. Here is an example. in Author response image 1, taken from the PhD thesis of Taly Ben-Shitrit, a PhD student in the lab.

**Author response image 1. sa4fig1:** 

What about cdc13 or ten1? Epistatic?

We did not test telomere length in combination with Ten1. Combining elg1Δ with cdc13-50 resulted in synergistic elongation. Given the complex genetic relationship between Stn1/Ten1 and Cdc13, it is hard to interpret this result.

Seems tenuous to use Y2H to decipher protein-protein interactions occurring out of context (i.e., not at telomere but at reporter gene promoter)

Y2H is a great method to detect interactions, even if they are transient. Whenever possible, we confirm our findings using co-IP or telo-ChIP.

Lines 268-270: It would be more accurate to state "can be" instead of "becomes" or "is" as they have not shown that SUMOylation or PCNA unloading have occurred.

We agree, and have changed the text.

Cdc13snm protein level?

Unfortunately our Western blot is not presentable, but the level of Cdc13snm was similar to that of the wt Cdc13, and this result has been already published by Hang et al., 2011.

Fig S3A: If SUMOylated Cdc13 mediates the Stn1-Elg1 interaction, why is Stn1-Elg1 interaction maintained in cdc13snm strain? This result seems to directly contradict the premise and overall conclusion of this section that Cdc13-SUMO mediates the (Y2H) interaction of Elg1 and Stn1.

According to our model, the interaction between Stn1 and Elg1 takes place upstream, and only then this complex interacts with SUMOylated Cdc13. Hence, if Cdc13 cannot be SUMOylated, the interaction Elg1-Stn1 is not lost, although Stn1 fails to interact with Cdc13, leading to a telomeric phenotype.

Line 279: which data establishes Stn1-Elg1 interaction as direct? Fig 2B co-Ip indicates physical but not necessarily direct interaction, but later the authors suggest that the interaction requires a SUMOylated intermediary, and Y2H in Fig. S3B doesn't demonstrate direct interaction.

We have changed the text, taking out the word "direct".

Co-Ip shows that interaction of Elg1 with Stn1 occurs mainly during later Sphase and with an overall delay compared to initial Elg1-Pol3 interaction.Co-IP Interaction between Cdc13 and Stn1 is reduced in the absence of Elg1The subsection title: "The interaction of Elg1 with Stn1 takes place at telomeres only at late S-phase" is not well supported by the data. I agree the data are consistent with the idea of the interactions occurring at telomeres but there's no direct evidence of this.

We have changed the subsection title. It now reads: " The interaction of Elg1 with Stn1 takes place only at late S-phase"

Model: Is unloading happening at the fork? Doesn't PCNA unloading have to follow its loading which occurred behind the fork particularly on the lagging strand? Model now suggest that Stn1 itself is SUMOylated.

Yes, according to the model Elg1 moves with the fork, unloading PCNA from the lagging strand. Once Elg1 reaches the telomeres, it interacts with Stn1 (Figure 5). This interaction requires SUMOylation of Stn1 or of some other protein, which is not PCNA (Figure 3D) nor Cdc13 (Figure S3A) and could be Stn1 itself or another telomeric protein (Hang et al., 2011)

Title is rather vague.

We think it summarizes what we present in the paper.

Abstract:"We report that SUMOylated PCNA acts as a signal that positively regulates telomerase activity."I don't think this is supported or a good description of what they find

Figure 1B clearly shows that SUMO-PCNA is both necessary and sufficient for telomere elongation.

"and dissected the mechanism by which Elg1 and Stn1 negatively regulates telomere elongation, coordinated by SUMO."Again, I don't think this is sufficiently supported and the model invokes SUMOylation events not demonstrated like Stn1, which might be a significant step forward.On the positive side, their model makes several predictions that they could test much more directly and rigorously: for example, examining the impact of the relevant mutations in the recruitment of proteins to the telomere.

We have dissected the mechanism, and future work will be devoted to examining the impact of the relevant mutations in the recruitment of proteins to the telomere.

**Reviewer #3 (Recommendations For The Authors):**
Comments:1. The telomere length analysis data presented here is consistent with an interpretation that Stn1 and Elg1 play roles in a similar telomere maintenance pathway because the telomere restriction fragment pattern in the double mutants are not longer than the stn1 single mutants. No comment is made with respect to the yellow bars in Figure 2 that presumably measure telomere length appearing to be slightly shorter than in the stn1 single mutants. It may be interesting and informative if the double mutants do in fact have some phenotype distinct from the single stn1 mutants. Is there an impact on viability in the double mutant?

Given the variable telomeric phenotype of the single stn1 mutants, slight variations in the measurement of the median telomere size are expected. The difference observed is not likely to be significant. What is important is that the double mutants with elg1Δ do not show longer telomeres. In terms of fitness, the stn1 mutants grow slightly slowly, but the elg1Δ mutation does not slow them down further.

1. It is somewhat surprising that no additional telomere length analysis is included that actually tests the proposed model, including whether this path could be operational only under certain conditions. Maybe this is a topic of the next paper?

Indeed, future work will explore the conditions under which PCNA SUMOylation is essential, and those under which is only needed.

1. Were the error bars in Figure 5F determined only from the experiment in E? Does this represent error in measuring the data from one biological replicate? The type of error should be made clear to avoid readers assuming the data represents measurements from more than one sample in more than one experiment. The data would be stronger if it represented measurements from multiple experiments.

The graph was made with data from three biological replicates. We show the best blot in Figure 5E. We have now stressed this in the Figure Legend.

1. Why was only one two hybrid reporter shown? Having the multiple reporters can give confidence in interactions. (Not a big deal here given the nice co-IP data.)

We thought that it is enough to show one reporter, as the results with a different reporter (B-gal assay) led to the same conclusions. since this did not add information and made the paper too lengthy (and boring), we took them out. In any case all data was verified by co-IP.

1. Line 414 - what are the 32P-radio labeled PCR fragments? Are these solely comprised of TG1-3 repeats of some length? A bit more detail in this aspect of the method could be helpful.

We have added an explanation on the probe in the Methods section.

1. Line 432-433 - which anti-HA or anti-My antibodies are these? (very minor detail)

We have added the details.